# MemLens: Uncovering Memorization in LLMs with Activation Trajectories

## Abstract

Large language models (LLMs) are commonly evaluated on challenging benchmarks such as AIME and Math500, which are susceptible to contamination and risk at being memorized. Existing detection methods, which primarily rely on surface-level lexical overlap and perplexity, demonstrate low generalization and degrade significantly when encountering implicitly contaminated data. In this paper, we propose **MemLens** (An Activation Lens for Memorization Detection) to detect memorization by analyzing the probability trajectories of numeric tokens during generation. Our method reveals that contaminated samples exhibit "shortcut" behaviors, locking onto an answer with high confidence in the model's early layers, whereas clean samples show more gradual evidence accumulation across the model's full depth. We observe that contaminated and clean samples exhibit distinct and well-separated reasoning trajectories. To further validate this, we inject carefully designed samples into the model through LoRA fine-tuning and observe the same trajectory patterns as in naturally contaminated data. These results provide strong evidence that MemLens captures genuine signals of memorization rather than spurious correlations.

## 1 Introduction

Large language models (LLMs) have achieved remarkable performance across reasoning, dialogue, and code generation tasks(Brown et al., 2020; OpenAI, 2023; Touvron et al., 2023). However, their success has raised growing concerns about *data contamination*, where benchmark test data (or their rephrased variants) appear in the training corpus. Such contamination can lead models to reproduce memorized solutions while appearing to reason, compromising claims of true generalization and raising privacy and intellectual property risks (Carlini et al., 2021; Zhao et al., 2024; Nasr et al., 2025). Contamination has been documented in widely used datasets such as MMLU and TruthfulQA (Deng et al., 2024). It can yield deceptively strong outputs: models may still solve compressed or even invalid problems, reflecting recall rather than reasoning (See Figure 1). Alternatively, Wu et al. (2025) demonstrate that reinforcement learning gains on math benchmarks can vanish once contaminated items are removed, underscoring the urgent need for systematic contamination detection.

Existing detection methods primarily rely on completion-based recall tests (Carlini et al., 2023; Nasr et al., 2025) and distributional probes such as perplexity and output agreement (Li et al., 2023a; Dong et al., 2024; Shi et al., 2024; Zhang et al., 2024b;a). Other approaches include adversarial compression (Schwarzschild et al., 2024) and partial probing (Zhao et al., 2024; Ye et al., 2024). While these methods provide valuable evidences of contamination, their reliance on surface lexical overlap limits robustness. When test problems are rephrased, paraphrased, or structurally altered, detection accuracy degrades substantially (Deng et al., 2024; Yao et al., 2024). For example, Yang et al. (2023) report that adding paraphrased MMLU items into training raises Llama 2 accuracy from 45–55% to nearly 89%, almost matching GPT-4. Such findings reveal that contamination persists even without verbatim overlap and motivate robust representation-level approaches.

We introduce **MemLens**, a detection algorithm leveraging activation trajectories to identify memorization footprints caused by data contamination. Our approach operates on the hypothesis that memorized samples induce "shortcut" reasoning paths within the model. Specifically, MemLens analyzes the evolution of probability distributions over numeric tokens across layers, tracking metrics like digit-specific probabilities, entropy, and maximum confidence. We observe that *con-*

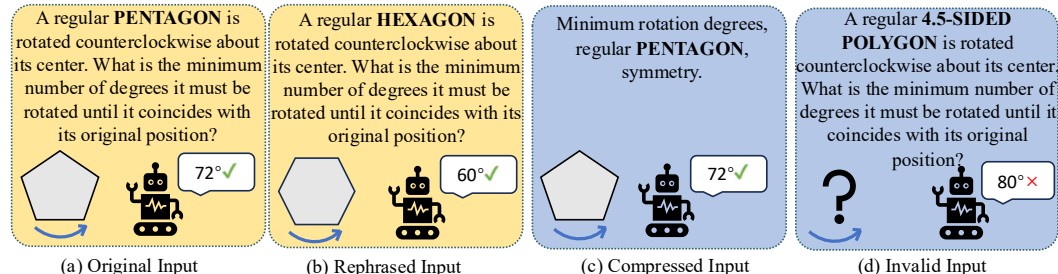

(a) Original Input  (b) Rephrased Input  (c) Compressed Input  (d) Invalid Input

**Figure 1:** These four examples illustrate the potential risk that memorization may masquerade as generalization. In (a), the model answers the original problem correctly, which is expected. In (b), after rephrasing, the model still provides the correct solution, appearing to generalize. In (c), however, even when the input is reduced to a minimal set of keywords, the model produces the correct answer together with a full chain-of-thought explanation. This behavior is unlikely to arise from genuine reasoning and instead strongly suggests direct recall of memorized content. In (d), the model attempts to generate an answer even for an invalid input, further underscoring its reliance on pattern matching rather than robust reasoning. Together, these cases reveal that model outputs may reflect hidden memorization signals disguised as generalization, a phenomenon that poses potential risks in downstream use; full chain-of-thought generations are provided in the Appendix B.

*taminated samples tend to lock onto an answer with high confidence much earlier in the forward pass*, whereas clean, unmemorized samples exhibit more gradual evidence accumulation that requires deeper processing. Unlike completion- or answer-based methods, MemLens remains robust under rephrasing, translation, and perturbation. Beyond detection performance, our analysis of activation dynamics reveals distinctive representation patterns of memorization, offering new insights into how contamination manifests inside models. To further establish causal validity, we conduct controlled LoRA injection experiments and show that injected samples induce the same trajectory signatures as naturally contaminated data. Our contributions are threefold: (i) a contamination detection algorithm that achieves robustness to rephrased inputs, (ii) representation-space analysis that characterizes memorization behaviors, and (iii) causal validation via LoRA injection, confirming that MemLens captures memorization rather than spurious signals.

## 2 PRELIMINARIES

### 2.1 MEMORIZATION DEFINITIONS

Let $\Sigma$ be the vocabulary and $\Sigma^\star$ the set of all finite sequences. A causal language model $M_\theta$ defines a conditional distribution $p_\theta(y \mid x)$ over continuations $y \in \Sigma^\star$ given a prompt $x \in \Sigma^\star$. We define the sequence-level log-likelihood as

$$s_\theta(y \mid x) \triangleq \sum_{t=1}^{|y|} \log p_\theta\big(y_t \mid x, y_{<t}\big). \tag{1}$$

A decoding functional $f_\theta^\pi(x)$ maps $x$ to a sequence according to a fixed policy $\pi$ (e.g., greedy decoding). Throughout, $y^\star \in \Sigma^\star$ denotes a specific training sample from the pretraining corpus.

**Extractable memorization.** We say that $y^\star$ is *extractably memorized* by $M_\theta$ if there exists a prompt $x \in \Sigma^\star$ and a decoding policy $\pi$ such that the decoded output $y$ equals $y^\star$ and has a nontrivial log-probability margin over any alternative continuation. Formally, there exist $x$ and margin $\kappa > 0$ such that

$$f_\theta^\pi(x) = y^\star; \quad s_\theta(y^\star \mid x) - \max_{y \neq y^\star} s_\theta(y \mid x) \geq \kappa. \tag{2}$$

This condition captures two aspects of extractability: (i) *verbatim reproduction* under a standard decoding rule, and (ii) a *separating margin* that rules out accidental ties due to sampling noise. In practice, one may upper-bound the second term by restricting $y$ to the $k$-best hypotheses or to a task-specific candidate set.

**Discoverable memorization.** We say that $y^\star = (y_{\text{pre}}, y_{\text{suf}})$ is *discoverably memorized* if there exists a split with a sufficiently informative prefix $y_{\text{pre}}$ such that the model (i) reproduces the exact suffix under length-constrained decoding, and (ii) assigns the suffix a per-token log-likelihood above a threshold. That is, there exist $y_{\text{pre}}, y_{\text{suf}}$ with $|y_{\text{pre}}| \geq m$ for some minimal prefix length $m$, and thresholds $\tau > 0$, $\kappa \geq 0$ such that

$$
\begin{aligned}
&\text{(i)} & f_\theta^{\pi,\,|y_{\text{suf}}|}(y_{\text{pre}}) &= y_{\text{suf}} \\
&\text{(ii)} & \tfrac{1}{|y_{\text{suf}}|}\, s_\theta(y_{\text{suf}} \mid y_{\text{pre}}) \geq \tau \quad \text{and} \quad s_\theta(y_{\text{suf}} \mid y_{\text{pre}}) - \max_{y \neq y_{\text{suf}}} s_\theta(y \mid y_{\text{pre}}) &\geq \kappa.
\end{aligned}
\tag{3}
$$

Clause (i) requires exact continuation under length-constrained decoding, which operationalizes "completion" tests, while Clause (ii) further ensures statistical preference for the suffix. The prefix length constraint $|y_{\text{pre}}| \geq m$ reflects the empirical observation that longer prefixes may be necessary to trigger recall. Unlike extractable memorization, discoverable memorization often requires such longer prefixes and may only surface under targeted probing (Carlini et al., 2023; Yang et al., 2023). These categories formalize memorization behaviors as model-level phenomena. In practice, contamination at the data level (i.e., benchmark items appearing in pretraining corpora) may manifest as either extractable or discoverable memorization during inference. However, recent findings show that such signals are fragile: rephrased or translated benchmark items can still induce contamination effects even without verbatim overlap (Yang et al., 2023; Deng et al., 2024). This motivates alternative strategies that capture subtler memorization dynamics.

## 2.2 REPRESENTATION PROBING

To analyze internal evidence relevant to contamination, we use the logit lens (Nostalgebraist, 2020) to project intermediate hidden states into the output space. Let $h_\ell \in \mathbb{R}^d$ denote the residual stream at layer $\ell$ and $W_{\text{LM}}$ the language modeling head. We define

$$
z_\ell = W_{\text{LM}} h_\ell, \qquad p_\ell = \text{softmax}(z_\ell).
\tag{4}
$$

This yields a layerwise predictive distribution $p_\ell$ over the vocabulary, allowing us to study how model preferences evolve with depth rather than only at the final output.

For tasks where only a subset of tokens is semantically relevant, we consider a token subset $S \subseteq \Sigma$ and the subset-normalized distribution

$$
p_\ell^{(S)}(t) = \frac{\exp(z_\ell[t])}{\sum_{u \in S} \exp(z_\ell[u])}, \qquad t \in S.
\tag{5}
$$

From $p_\ell^{(S)}$ one can further derive compact summaries such as entropy, maximum probability, or margins. These quantities capture how model preferences over $S$ evolve across layers and will serve as building blocks for our method in Section 3.

## 3 METHODOLOGY

In this section, we introduce the proposed **MemLens** (An Activation Lens for Memorization Detection) framework for identifying contamination in LLMs.

### 3.1 PROBLEM FORMULATION

We consider a LLM $M$ that takes a textual problem $x$ as input and produces an answer $a$. Our objective is to determine whether $(x, a)$ is *contaminated* (i.e., the model has memorized this pair during training or fine-tuning) or *clean* (i.e., not memorized). Formally, we construct a binary classification problem from a dataset $\{(x_i, a_i, y_i)\}_{i=1}^N$, where $y_i \in \{0, 1\}$ indicates whether the sample is contaminated ($y_i = 1$) or clean ($y_i = 0$).

This formulation allows us to design a representation-based discriminator $f_\theta$ that takes intermediate activations from $M$ as input and predicts the likelihood of contamination.

## 3.2 REPRESENTATION EXTRACTION

Following the logit lens framework, we probe the hidden states of $M$. Let $h_\ell \in \mathbb{R}^d$ denote the residual stream activation at layer $\ell$. Applying the output embedding $W_{\text{LM}}$ yields logits and the probability distribution over the vocabulary:

$$z_\ell = h_\ell W_{\text{LM}}, \quad p_\ell = \text{softmax}(z_\ell). \tag{6}$$

Since we focus on numerical reasoning tasks, we restrict the distribution to digit tokens. Let $\mathcal{V}_d$ denote the set of token IDs corresponding to digits $d \in \{0, \ldots, 9\}$. The digit-only probability at layer $\ell$ is defined as:

$$p_\ell(d) = \frac{\sum_{t \in \mathcal{V}_d} p_\ell(t)}{\sum_{j=0}^{9} \sum_{t \in \mathcal{V}_j} p_\ell(t)}. \tag{7}$$

We further compute two scalar statistics: digit entropy $H_\ell$ and maximum confidence $m_\ell$ as follows:

$$H_\ell = -\sum_{d=0}^{9} p_\ell(d) \log p_\ell(d), \tag{8}$$

$$m_\ell = \max_d p_\ell(d). \tag{9}$$

## 3.3 FEATURE CONSTRUCTION

For each sample $(x, a)$, we identify the first generation position corresponding to the answer and extract trajectories across the first $T$ layers. The feature vector at layer $\ell$ is constructed by concatenating the digit probabilities, entropy, max confidence, and their first-order temporal differences:

$$X = \left[ p_\ell(0..9), \ H_\ell, \ m_\ell, \ \Delta p_\ell, \ \Delta H_\ell, \ \Delta m_\ell \right]_{\ell=1}^{T}, \tag{10}$$

where $\Delta$ denotes first-order temporal differences, e.g., $\Delta H_\ell = H_\ell - H_{\ell-1}$. This results in a multichannel sequence representation $X \in \mathbb{R}^{C \times T}$, with $C = 24$ channels (12 raw features and 12 difference features).

Our feature design was guided by an empirical investigation. We initially hypothesized that contaminated samples would simply "lock onto" the correct digit with high confidence much earlier in the forward pass. However, we found this was an oversimplification, as multiple digits often remain active across layers even for clean samples. This led to our core insight: the signal of memorization is not a single, early decision, but rather **the entire activation pattern itself**, i.e., how the probabilities of all digits evolve, compete, and resolve across the model's depth. Rather than hand-crafting rules to capture this complex dynamic, we construct features that represent the full trajectory and defer to a learned discriminator to find the distinguishing signals. This design directly tests whether these internal trajectories contain a reliable and detectable footprint of memorization.

## 3.4 CNN-BASED DISCRIMINATOR

We train a lightweight 1D convolutional neural network with hidden dimension $d = 128$ to classify whether a trajectory $X$ originates from a contaminated or clean sample. The network consists of three convolutional layers with ReLU activation and a global average pooling layer. The classifier produces logits

$$z = \text{CNN}(X) \in \mathbb{R}^d, \quad o = Wz + b \in \mathbb{R}^2, \quad \hat{y} = \text{softmax}(o). \tag{11}$$

The training objective is the standard cross-entropy loss, where $y_{i,c}$ is the one-hot ground-truth label:

$$\mathcal{L}(\theta) = -\sum_{i=1}^{N} \sum_{c=1}^{2} y_{i,c} \log \hat{y}_{i,c}. \tag{12}$$

Importantly, the discriminator addresses our central question: if contamination induces distinctive memorization footprints in activation trajectories, the classifier should be able to separate contaminated from clean samples based on these features.

**Table 1:** Performance comparison of different methods across four text distributions. All numerical results are reported to one decimal place. Cont. means contaminated.

| | Original | | Rephrased | | Translated | | Perturbed | |
|---|---|---|---|---|---|---|---|---|
| | Cont. ↑ | Clean ↓ | Cont. ↑ | Clean ↓ | Cont. ↑ | Clean ↓ | Cont. ↑ | Clean ↓ |
| **Qwen2.5-7B** | | | | | | | | |
| Completion-based | 100.0 | 11.6 | 16.7 | 3.7 | 12.2 | 2.6 | 15.8 | 3.0 |
| TS-Guessing | 92.7 | 14.5 | 21.3 | 15.0 | 3.6 | 0.0 | 17.7 | 14.6 |
| Perplexity | 87.6 | 1.1 | 29.2 | 0.4 | 10.8 | 0.7 | 19.0 | 0.0 |
| Output Distribution | 64.7 | 28.0 | 62.6 | 45.3 | 26.4 | 16.5 | 58.9 | 39.3 |
| MemLens (Ours) | 95.6 | 5.6 | 63.1 | 2.2 | 20.0 | 1.5 | 60.1 | 1.5 |
| **Qwen2.5-Math-7B** | | | | | | | | |
| Completion-based | 100.0 | 8.7 | 18.3 | 3.7 | 13.6 | 2.6 | 15.4 | 1.9 |
| TS-Guessing | 96.4 | 11.4 | 19.5 | 13.9 | 1.5 | 0.0 | 17.1 | 12.4 |
| Perplexity | 96.1 | 3.8 | 38.4 | 3.7 | 30.9 | 3.0 | 26.7 | 3.0 |
| Output Distribution | 71.3 | 19.3 | 50.5 | 20.9 | 17.8 | 11.2 | 50.5 | 26.6 |
| MemLens (Ours) | 89.4 | 5.6 | 56.2 | 2.6 | 64.2 | 9.4 | 51.7 | 1.9 |
| **Llama3.1-8B** | | | | | | | | |
| Completion-based | 100.0 | 14.4 | 4.3 | 6.3 | 3.2 | 3.4 | 2.9 | 4.8 |
| TS-Guessing | 98.2 | 17.3 | 29.1 | 0.3 | 0.3 | 0.2 | 0.3 | 0.2 |
| Perplexity | 93.7 | 11.2 | 36.1 | 10.6 | 31.5 | 8.7 | 24.9 | 8.1 |
| Output Distribution | 60.0 | 18.0 | 59.8 | 36.4 | 39.3 | 20.2 | 62.8 | 34.8 |
| MemLens (Ours) | 95.7 | 12.8 | 63.0 | 4.1 | 46.4 | 4.5 | 61.0 | 4.5 |
| **Qwen-3-8B** | | | | | | | | |
| Completion-based | 100.0 | 14.2 | 19.4 | 2.6 | 13.8 | 2.6 | 18.8 | 1.9 |
| TS-Guessing | 91.3 | 21.7 | 20.1 | 19.5 | 3.8 | 1.9 | 19.0 | 17.6 |
| Perplexity | 70.0 | 39.7 | 63.7 | 42.3 | 84.4 | 64.8 | 57.5 | 36.7 |
| Output Distribution | 78.0 | 33.0 | 83.8 | 33.3 | 33.1 | 19.1 | 64.4 | 32.9 |
| MemLens (Ours) | 87.5 | 14.8 | 93.1 | 13.1 | 78.1 | 10.1 | 64.4 | 13.9 |

## 4 EXPERIMENTS

In this section, we present experiments to demonstrate the effectiveness of **MemLens** on memorization detection. Our experiments are designed to address the following research questions:

- **RQ1:** Do activation trajectories yield reliable signals for distinguishing memorized from clean samples, and are these signals robust to input variations?
- **RQ2:** Does fine-tuning on clean data causally increase the detector's memorization scores, thereby providing evidence that the detector captures memorization signals?
- **RQ3:** Where are memorization signals encoded within the model, and how well does our detector generalize across different evaluation settings?

### 4.1 EXPERIMENTAL SETUP

**Models and Datasets.** We evaluate four open-source LLMs including Qwen2.5-7B, Qwen2.5-Math-7B, Qwen3-8B, and Llama-3.1-8B. For models in Qwen family, contaminated samples are constructed from *AIME 1983–2024* and *Math500* by prompting the model and collecting completions that exactly match the gold answer. Clean samples are constructed from *AIME 2025*, *Minerva Math*, and *LiveMathBench*, which were released after these models. For Llama-3.1-8B, contaminated samples are constructed from *AIME 1983–2024*, while *Math500* is used as clean data along with *AIME 2025*, *Minerva Math*, and *LiveMathBench*, since *Math500* was released later. To evaluate robustness, we additionally introduce three variants: *Rephrased* (paraphrased wording), *Perturbed* (surface-level edits), and *Translated* (French translation), by prompting GPT-4o-mini. We report the prompt details in Appendix C.

**Baselines.** We compare against four prior approaches.(i) *Completion-based* (Carlini et al., 2023): a sample is flagged as contaminated if the model's greedy completion exactly matches the refer-

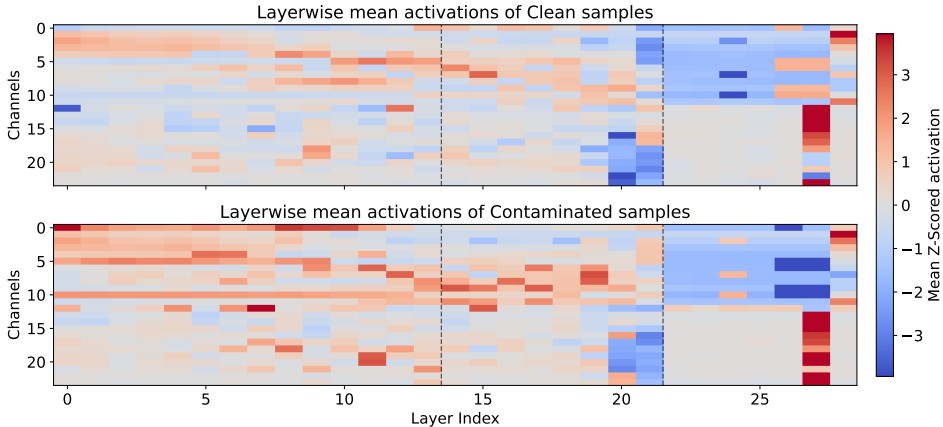

**Figure 2:** Class-mean activation heatmaps. Average channel activations across layers for clean vs. contaminated samples. Red regions denote stronger activation and blue denotes weaker activation.

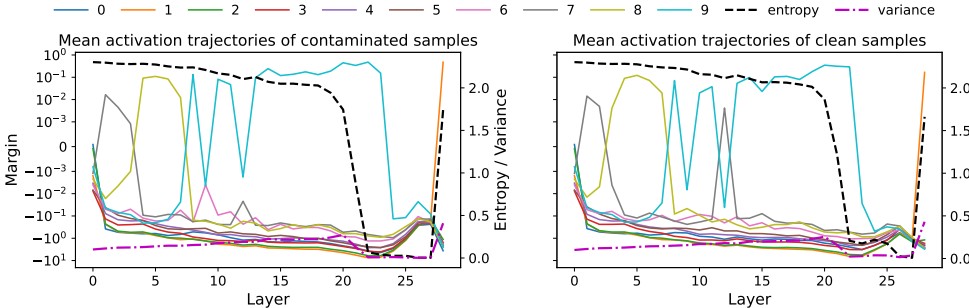

**Figure 3:** Mean activation trajectories of correctly classified samples. Each panel shows layerwise dynamics at the answer position for contaminated (left) and clean (right) samples.

ence answer. (ii) *TS-Guessing* (Deng et al., 2024): contamination is detected if the model can infer the gold answer through slot-filling or symbolic transformations, without requiring exact string match. These two belong to completion-driven baselines. (iii) *Perplexity* (Li et al., 2023b): computes the log-likelihood of the reference answer under the model; unusually low perplexity is taken as evidence of memorization. (iv) *Output Distribution* (Dong et al., 2024): measures the similarity between greedy decoding and multiple stochastic samples, higher similarity indicates stronger memorization. These two belong to distribution-based baselines.

**Metrics.** We adopt four evaluation metrics.(i) *Exact Match (EM / pass@1)*: the fraction of generations exactly matching the gold answer, measuring strict correctness. (ii) *Rouge-L*: longest common subsequence between prediction and reference, capturing partial overlap. (iii) *Youden's J index*: $J = \text{TPR} - \text{FPR}$, used to select the decision threshold that balances sensitivity and specificity. (iv) *Contamination Detection Rate*: the proportion of samples flagged as contaminated by our detector at the chosen threshold, quantifying how many suspected memorized cases are detected.

## 4.2 MAIN RESULTS

**Quantative Analysis.** We evaluate all methods on the four text distributions: original, rephrased, translated, and perturbed. The results shows clear differences in Table 1. Completion-based approaches (Completion-based and TS-Guessing) almost entirely rely on whether the model can verbatim reproduce training text: they achieve nearly 100% detection on original problems, but collapse once the inputs are rephrased or perturbed, with detection rates dropping below 20%. Output-distribution methods (Perplexity and Output Distribution) exploit probability signals and are somewhat more robust, but remain unstable: Perplexity produces very high false positives on clean sam-

**Table 2:** Detection results under LoRA fine-tuning with varying rank $r$. For the pre-trained model, all samples are classified as clean. As $r$ increases, the Predicted Contamination Rate (PCR) rises before dropping at very large $r$. EM, RougeL and PCR socres are reported as percentages.

| Model | Trainable-parameters | EM (pass@1) | RougeL | PCR |
|---|---|---|---|---|
| Pre-trained Model | 0 | 17.8 | 35.4 | 2.2 |
| *Fine-tuned Model* | | | | |
| $r = 8$ (32 epochs) | 20M | 27.2 | 53.0 | 14.9 |
| $r = 16$ (32 epochs) | 40M | 41.8 | 65.5 | 19.0 |
| $r = 32$ (32 epochs) | 80M | 52.2 | 73.6 | 29.7 |
| $r = 64$ (32 epochs) | 170M | 58.9 | 77.6 | 38.0 |
| $r = 128$ (32 epochs) | 340M | 69.8 | 81.6 | 39.9 |
| $r = 256$ (32 epochs) | 670M | 75.9 | 88.1 | 44.3 |
| $r = 512$ (32 epochs) | 1.3B | 78.5 | 89.1 | 45.1 |

ples in some models, while Output Distribution fluctuates widely across distributions. In contrast, our MemLens detector is consistently stable across all models and variants: it achieves significantly higher detection rates on contaminated samples while keeping the lowest false positive rates on clean ones. For example, on rephrased Qwen2.5-7B data, completion-based methods detect only 16.7% of contaminated cases, whereas our detector achieves 63.1%; on perturbed Llama3.1-8B data, Output Distribution mislabels over 30% of clean cases, while our detector limits this to just 4.5%. These results indicate that baseline methods either rely on superficial cues or yield unstable signals, whereas our approach captures deeper and more reliable activation patterns within the model.

**Representation Analysis.** We further examine the internal activation patterns to understand how our detector distinguishes contaminated from clean samples. We observe that contaminated samples tend to show stronger and more concentrated activations in earlier layers, whereas clean samples remain more diffuse and only strengthen in later layers (see Figure 2). We also track the average activation trajectories across all correctly classified examples in Figure 3. For contaminated inputs, margin divergence and early dominance of a single digit channel emerge within the first ten layers, indicating premature overconfidence; for clean inputs, evidence accumulates more gradually and uncertainty stays higher for longer. Together, these results reveal that contaminated data consistently induce shortcut-like activations that are earlier and more concentrated, whereas clean data require deeper and more distributed processing before a decision is made.

## 4.3 LoRA Injection Evaluation

To further validate the effectiveness of our method, we conduct *LoRA injection experiments*. Specifically, we fine-tune the base model $M_{base}$ on a small set of clean samples using LoRA with varying rank $r$, obtaining $M_{lora}$. Using the same evaluation dataset, we extract features from both models and apply our trained discriminator $f_\theta$. Since the fine-tuning data are exactly from clean samples, increasing predictions from $M_{base}$ to $M_{lora}$ provides causal evidence that $f_\theta$ captures genuine memorization signals.

As shown in Table 2, for $M_{base}$, these samples are almost always classified as clean (contaminated rate 2.2%). However, after LoRA fine-tuning, the a larger fraction of samples are identified as contaminated, with the rate increasing to 44.3% at $r = 256$ and 45.1% at $r = 512$. This upward trend at small-to-moderate $r$ is also aligned with task performance: both EM (pass@1) and Rouge-L steadily improve as $r$ grows, indicating that the model is leveraging injected memorization to answer correctly. Overall, the parallel increase of task accuracy and detected contaminated rate at reasonable ranks demonstrates that memorization injected by LoRA is consistently captured by our method, verifying that the learned signal reflects true causal memorization rather than spurious correlations. In Figure 4, we present a controlled case study tracing a single evaluation sample across LoRA ranks. With the detector $f_\theta$ and threshold fixed, the predicted contamination probability increases systematically from 0.20 (base) to 0.62 ($r$=16), 0.92 ($r$=64), and 0.98 ($r$=256). Since neither the discriminator, prompts, nor test data change, this monotonic increase cannot be explained by noise or spurious correlations. It must arise from the injected exposures, providing direct causal evidence that the detector is sensitive to memorization recall. The heatmaps illustrate how this signal emerges:

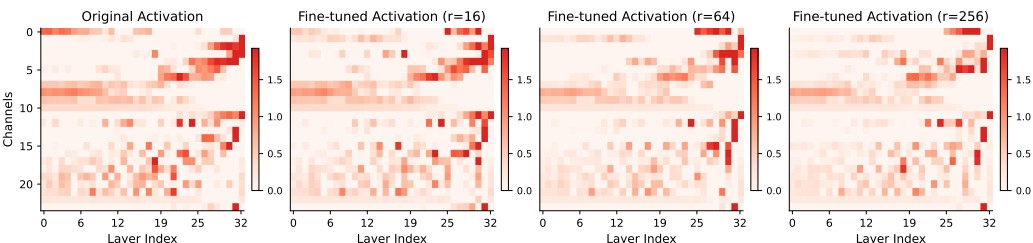

**Figure 4:** Layer-wise activations for one sample across LoRA ranks, with layers on the x-axis, 24 feature channels on the y-axis, and color showing normalized values.

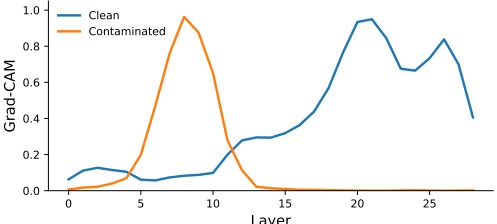

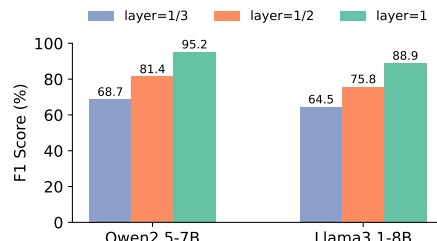

**Figure 5:** Class-mean Grad-CAM visualization comparing clean vs. contaminated samples.

**Figure 6:** F1 scores across different layers.

in the base model, digit channels from 1 to 9 remain diffuse in the early layers without a clear preference. After LoRA fine-tuning, however, one channel begins to dominate in the early layers, and this dominance becomes more pronounced as $r$ grows. This progression mirrors the broader trend in Figure 3, where contaminated samples exhibit earlier and stronger channel commitment, even when the chosen digit does not correspond to the correct answer.

### 4.4 ABLATION STUDY

**Layer Selection.** In the main experiments, all results are based on discriminators trained on activations from the full model depth. Since earlier analyses indicated that early layers provide especially strong signals for contamination detection, we conducted an ablation study restricting the discriminator to activations from only the first one-third or the first half of the layers. As shown in Figure 6, using only the first one-third of the layers already delivers meaningful performance, with F1 scores of 68.7% on Qwen2.5-7B and 64.5% on Llama3.1-8B. Using the first half of the layers improves results further, reaching 81.4% and 75.8% respectively. The best performance is consistently obtained when the full depth is used, yielding 95.2% on Qwen2.5-7B and 88.9% on Llama3.1-8B. These results show that early layers already encode strong memorization cues, but aggregating signals across the full depth provides the most reliable detection. This quantitative trend is further supported by Grad-CAM (Selvaraju et al., 2017) visualizations (Figure 5), where contaminated samples exhibit sharp importance peaks in shallow layers, while clean samples show delayed and smoother distributions. Together, these results confirm that early layers encode strong memorization cues, even though aggregating signals across the full depth provides the most reliable detection.

**Evaluation on MMLU.** A key question is whether our detector can expose meaningful structure in a widely used benchmark where the true contamination status is unknown. We refer to samples that the detector predicts as contaminated as *seen-like*, and to those predicted as clean as *not-seen*. These labels are used only for grouping by detector output, rather than for making definitive claims about whether a sample is truly contaminated. We apply the detector trained on *Qwen2.5-Math-7B* to the *mathematics subset of MMLU* in a zero-shot setting. As shown in Figure 7, the detector labels 94.9% of the problems as *seen-like* and only 5.1% as *not-seen*. This split is not arbitrary: *Qwen2.5-Math-7B* model achieve markedly higher performance on the *seen-like* group, with Top-1 accuracy higher by 0.33 and MRR@6 higher by 0.23. Confidence intervals confirm these gaps are statistically significant. These results indicate that a substantial portion of the model's success on

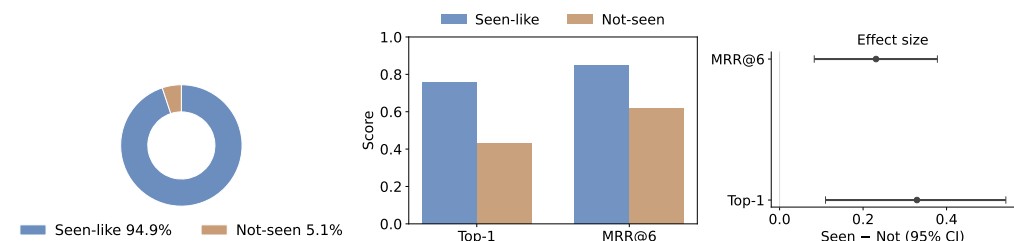

**Figure 7:** Dataset composition and group-level performance comparison. **Left:** fraction of samples classified as seen-like (94.9%) versus not-seen (5.1%). **Middle:** Top-1 accuracy and MRR@6 for the two groups. **Right:** effect sizes (Seen–Not) with 95% confidence intervals.

MMLU mathematics may derive from recalling previously seen problems rather than solving unseen ones from first principles. Our detector thus recovers hidden memorization signals without accessing answers, offering a principled way to reassess what current benchmark performance actually reflects.

## 5 RELATED WORK

**Contamination and Copyright.** Benchmark results are often inflated when training and test overlap. Leakage has been documented in MMLU and other benchmarks (Zhou et al., 2023; Deng et al., 2024), while rephrased and translated variants can raise accuracy without verbatim overlap (Yang et al., 2023; Yao et al., 2024). Evaluation-side defenses attempt to mitigate these risks: CLEAN-EVAL and inference-time filtering sanitize benchmarks (Zhu et al., 2024b;a), DCQ provides automated auditing (Golchin & Surdeanu, 2023), and task-level contamination undermines few-shot claims (Li & Flanigan, 2023). A practical dimension is copyright. DE-COP identifies copyrighted content in pretraining corpora (Duarte et al., 2024), audits reveal explicit copyright violations in model outputs (Karamolegkou et al., 2023), partial probing quantifies exposure risk (Zhao et al., 2024), and further tests examine whether models comply with copyright notices or reproduce licensed code (Xu et al., 2024; Yu et al., 2023). Such findings underscore that contamination is not only a technical artifact but also a legal and ethical risk: reproducing protected material can mislead benchmark evaluations and at the same time threaten compliance with intellectual property.

**Detection and Mechanisms.** Detection protocols formalize memorization as extractable or discoverable and extend to compression-based probing with shorter triggers (Carlini et al., 2023; Nasr et al., 2025; Schwarzschild et al., 2024), which capture verbatim recall yet still underestimate hidden forms under longer or rephrased inputs. Statistical calibration methods broaden coverage: minimum-probability detectors (Shi et al., 2024) and divergence-based scoring (Zhang et al., 2024b) flag anomalous likelihoods, paired confidence tests improve statistical reliability (Zhang et al., 2024a), and black-box calibration extends applicability to proprietary models (Ye et al., 2024). Interventions provide stronger causal evidence, with LoRA injection showing how added data alters detection signals (Tu et al., 2024) and temporal audits tracing when benchmarks leaked into training corpora (Golchin & Surdeanu, 2024). Beyond detection, boundary studies interrogate whether high performance reflects reasoning or memorization, through idiom understanding (Kim et al., 2025), knowledge updating (Li & Goyal, 2025), generalization failures (Dong et al., 2024), and overestimation in machine translation (Kocyigit et al., 2025). Together, these studies suggest that while existing methods advance coverage and robustness, they remain tied to surface cues or distributional anomalies, motivating representation-level analysis to directly expose memorization dynamics.

## 6 CONCLUSION

We introduce MemLens, a representation-based framework for detecting memorization in LLMs. By analyzing activation trajectories, MemLens provides reliable signals that remain stable under rephrasing and perturbation. Our comprehensive experiments, including controlled LoRA injections, not only confirm the robustness of this approach but also provide strong causal evidence that these activation patterns are a genuine footprint of memorization rather than spurious correlations.

## REPRODUCIBILITY STATEMENT

We make substantial efforts to ensure the reproducibility of our results. All datasets and preprocessing pipelines are described in Section 4.1, and Appendix C. Implementation details of the classifier training, feature extraction, and LoRA fine-tuning experiments are provided in Appendix E.

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

## A  THE USE OF LARGE LANGUAGE MODELS

During the preparation of this paper, we made limited use of large language models (LLMs) to assist with writing. Specifically, we employed them to refine the phrasing of certain sentences and to discuss the appropriateness of particular word choices.

## B  FULL CoT OUTPUTS

---

**Case Study 1: Original Question (Baseline)**

**Question:**
A regular pentagon is rotated counterclockwise about its center. What is the minimum number of degrees it must be rotated until it coincides with its original position?

---

**Model Output:**
72 degrees.
To determine the minimum number of degrees a regular pentagon must be rotated counterclockwise about its center until it coincides with its original position, we need to consider the symmetry of the pentagon. A regular pentagon has rotational symmetry, meaning it looks the same after being rotated by certain angles around its center.
The order of rotational symmetry for a regular pentagon is 5. This means there are 5 different positions where the pentagon looks exactly the same when rotated. The angle of rotation for each symmetry position is given by dividing 360 degrees (a full rotation) by the order of symmetry.
So, we calculate:

$$\frac{360°}{5} = 72°$$

Therefore, the minimum number of degrees the pentagon must be rotated counterclockwise until it coincides with its original position is $\boxed{72}$.

---

**Table 3:** Case study 1: Original baseline question.

---

**Case Study 2: Core Number Changed (Hexagon)**

**Question:**
A regular hexagon is rotated counterclockwise about its center. What is the minimum number of degrees it must be rotated until it coincides with its original position?

---

**Model Output:**
60 degrees.
To determine the minimum number of degrees a regular hexagon must be rotated counterclockwise about its center until it coincides with its original position, we need to consider the symmetry of the hexagon. A regular hexagon has rotational symmetry, meaning it looks the same after being rotated by certain angles around its center.
A regular hexagon has 6 equal sides and 6 equal angles. The sum of the interior angles of a hexagon is given by the formula $(n - 2) \times 180°$, where $n$ is the number of sides. For a hexagon ($n = 6$), the sum of the interior angles is:

$$(6 - 2) \times 180° = 4 \times 180° = 720°$$

However, we are interested in the angles of rotation that will map the hexagon onto itself. Since the hexagon has 6 equal sides, it can be divided into 6 equal parts. Therefore, the angle of rotation that will map the hexagon onto itself is:

$$\frac{360°}{6} = 60°$$

This means that if the hexagon is rotated by 60 degrees counterclockwise, it will coincide with its original position. any multiple of 60 degrees (i.e., 120 degrees, 180 degrees, 240 degrees, 300 degrees, 360 degrees, etc.) will also map the hexagon onto itself, but the minimum positive angle is 60 degrees.
Thus, the minimum number of degrees the hexagon must be rotated until it coincides with its original position is $\boxed{60}$.

---

**Table 4:** Case study 2: Hexagon input, model outputs correct result.

---

**Case Study 3: Keyword Summary Input**

---

**Question:**
minimum rotation degrees, regular pentagon, symmetry

---

**Model Output:**
$72°$
The minimum rotation degrees that will map a regular pentagon onto itself is $72°$. This is because a regular pentagon has five sides, and a full rotation of $360°$ divided by five gives $72°$. This means that rotating the pentagon by $72°$ will bring it back to a position where it looks exactly the same as it did before the rotation.

To understand this better, let's consider the symmetry of a regular pentagon. A regular pentagon has five lines of symmetry, each passing through a vertex and the midpoint of the opposite side. When the pentagon is rotated by $72°$, each vertex moves to the position of the next vertex in a clockwise or counterclockwise direction. This rotation aligns the pentagon with itself, leaving it unchanged.

Therefore, the minimum rotation degrees that will map a regular pentagon onto itself is $\boxed{72°}$.

---

**Table 5:** Case study 3: Keyword-based input, model still outputs the correct reasoning.

---

**Case Study 4: Input Floating Number (4.5)**

---

**Question:**
A regular 4.5-sided polygon is rotated counterclockwise about its center. What is the minimum number of degrees it must be rotated until it coincides with its original position?

---

**Model Output:**
80 degrees.
To determine the minimum number of degrees a regular 4.5-sided polygon must be rotated until it coincides with its original position, we start by understanding the properties of regular polygons and their rotational symmetry.

A regular polygon with $n$ sides can be rotated by $\frac{360°}{n}$ degrees to coincide with its original position. This is because a full rotation of $360°$ degrees will bring the polygon back to its starting position, and dividing this by the number of sides gives the smallest angle of rotation that will result in the polygon coinciding with itself.

In this problem, the polygon has 4.5 sides. However, a polygon must have an integer number of sides. Therefore, we need to interpret the problem as referring to a polygon with a fractional number of sides in a theoretical or conceptual sense, and we can use the same formula to find the angle of rotation.

The formula for the angle of rotation is:

$$\text{Angle of rotation} = \frac{360°}{4.5}$$

To simplify this, we can convert 4.5 to a fraction:

$$4.5 = \frac{9}{2}$$

Now, substitute this into the formula:

$$\text{Angle of rotation} = \frac{360°}{\frac{9}{2}} = 360° \times \frac{2}{9} = \frac{360° \times 2}{9} = \frac{720°}{9} = 80°$$

Thus, the minimum number of degrees the polygon must be rotated until it coincides with its original position is:

$$\boxed{80}$$

---

**Table 6:** Case study 4: Input with non-integer polygon sides.

# C    REPHRASE DATASET PROMPT

## C.1    PROBLEM TRANSFORMATION SETUP

We applied controlled prompting to generate different variants of math problems for our experiments.    All generations were obtained using the `GPT-4o-mini` model (via the `chat.completions` API). To ensure variability while preserving mathematical correctness, we set the sampling temperature to 0.7.

We designed three transformation tasks:

- **Rephrasing**: synonym replacement or structure change without altering semantics.
- **Perturbation**: minor linguistic noise while keeping numbers/LaTeX fixed.
- **Translation (French)**: translate only the surrounding text, keeping numbers/LaTeX unchanged.

The exact prompts used for each task are shown in Table 7.

| Task | Prompt Template |
|------|-----------------|
| Rephrase | Rephrase the following math problem in English. Requirements:
1. Keep ALL numbers and LaTeX expressions exactly the same.
2. Do not change the mathematical meaning.
3. Output ONLY the rephrased problem, no explanations.
Problem: {q} |
| Perturb | Rewrite the following math problem with minor linguistic perturbations.  You may change word order, replace synonyms, or alter sentence structure. Requirements:
1. Keep ALL numbers and LaTeX expressions exactly unchanged.
2. Preserve the exact mathematical meaning.
3. Output ONLY the perturbed problem.
Problem: {q} |
| Translate (French) | Translate the following math problem into French. Requirements:
1. Keep ALL numbers and LaTeX expressions unchanged.
2. Translate only the surrounding text.
3. Output ONLY the translated problem, no explanations.
Problem: {q} |

**Table 7:** Prompt templates used to generate rephrased, perturbed, and translated math problems. All generations used GPT-4o-mini with temperature $= 0.7$.

# D MORE ACTIVATION TRAJECTORIES ACROSS MODELS

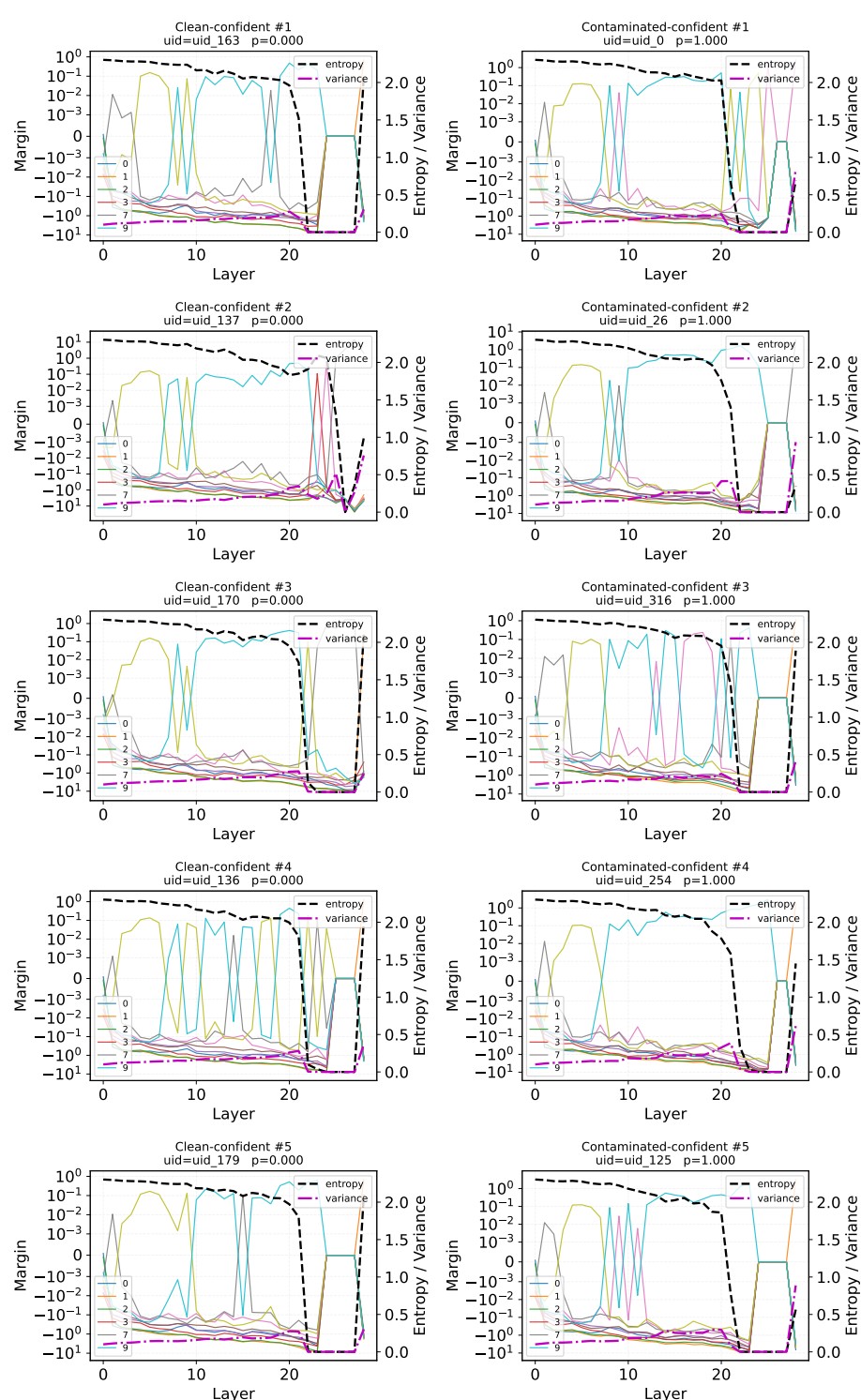

**Figure 8:** Activation trajectories of the 10 most confidently classified samples by the discriminator trained on Qwen2.5-7B. Clean samples are shown on the left, while contaminated samples are shown on the right.

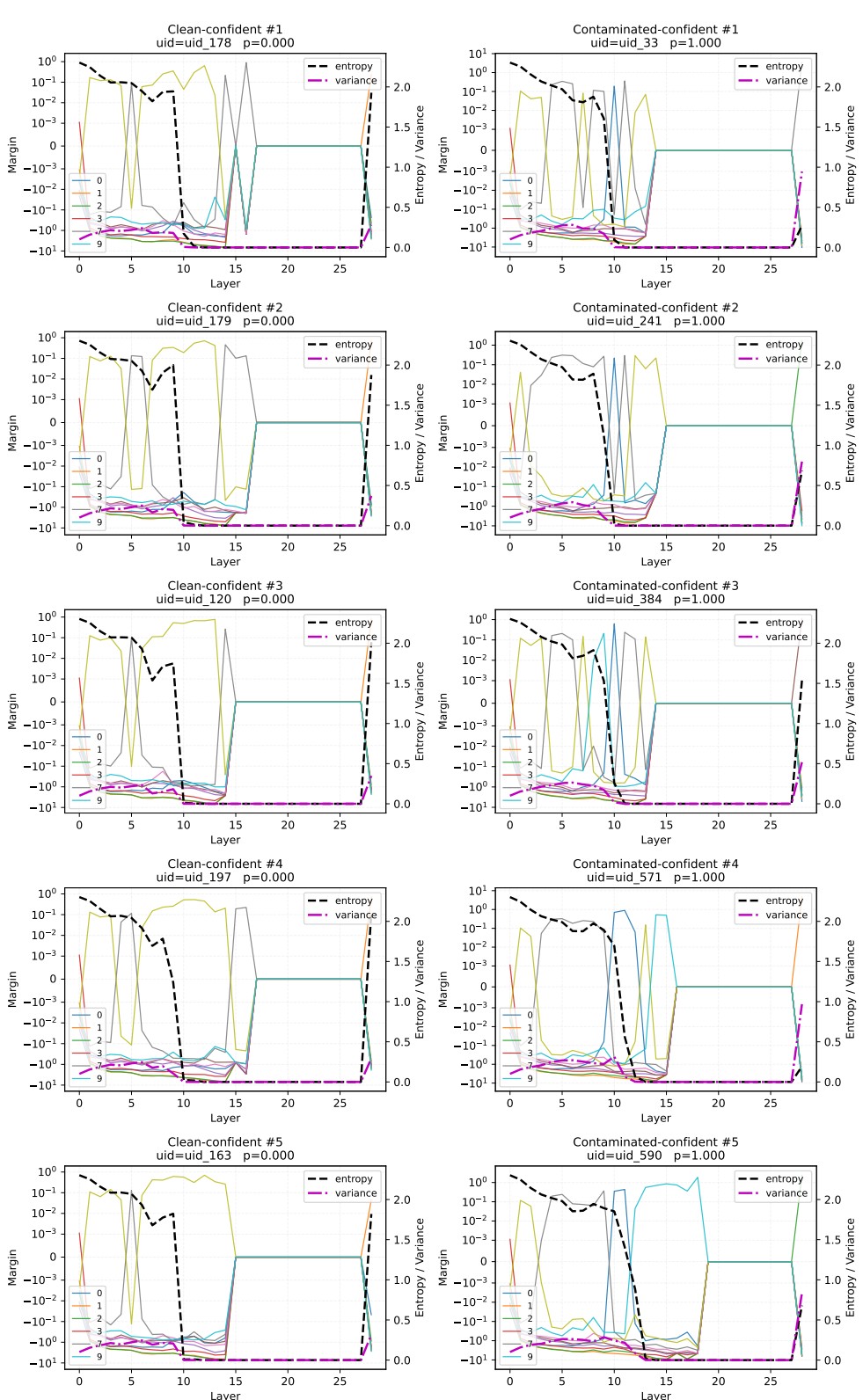

**Figure 9:** Activation trajectories of the 10 most confidently classified samples by the discriminator trained on Qwen2.5-Math-7B. Clean samples are shown on the left, while contaminated samples are shown on the right.

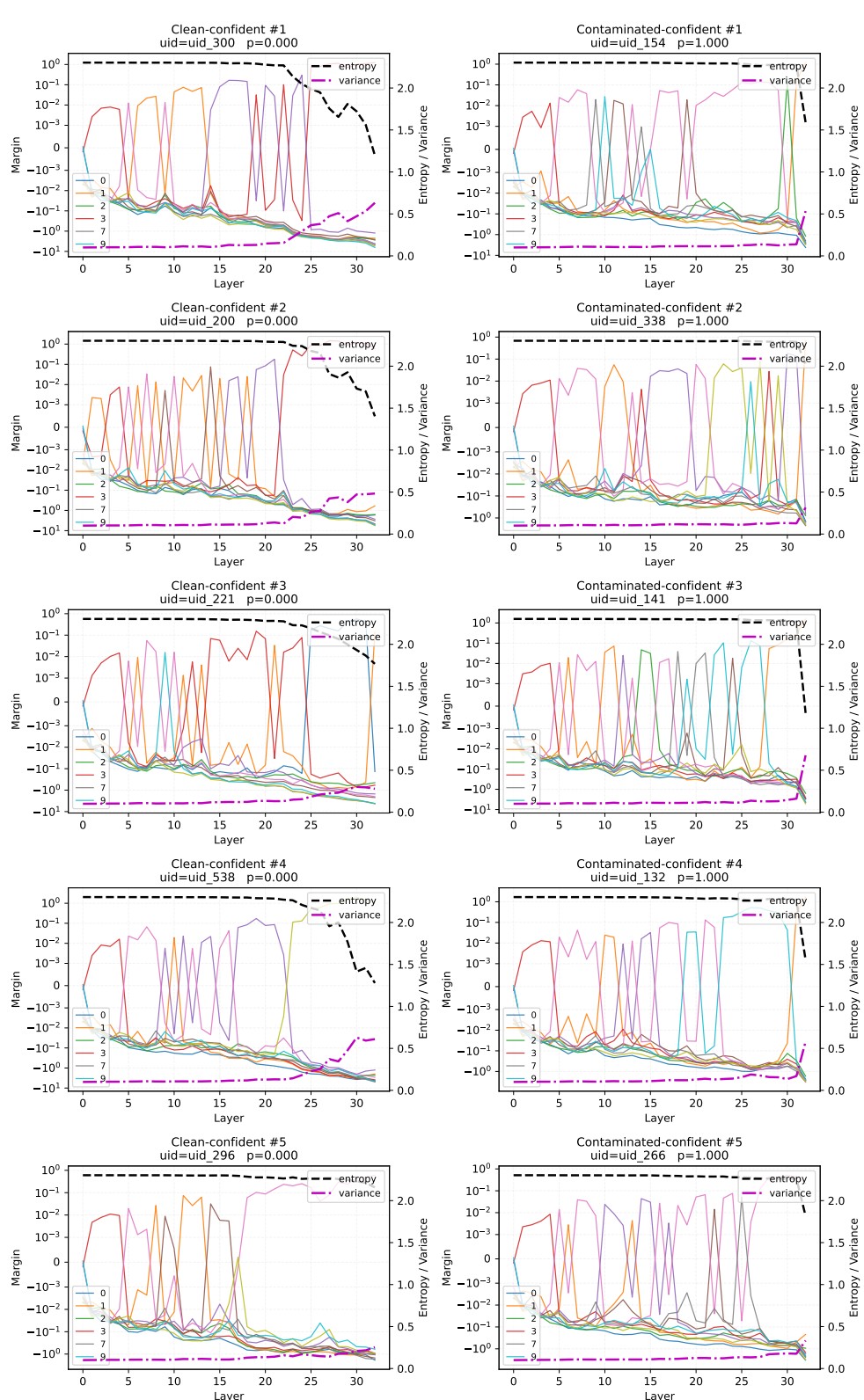

**Figure 10:** Activation trajectories of the 10 most confidently classified samples by the discriminator trained on Llama3.1-8B. Clean samples are shown on the left, while contaminated samples are shown on the right.

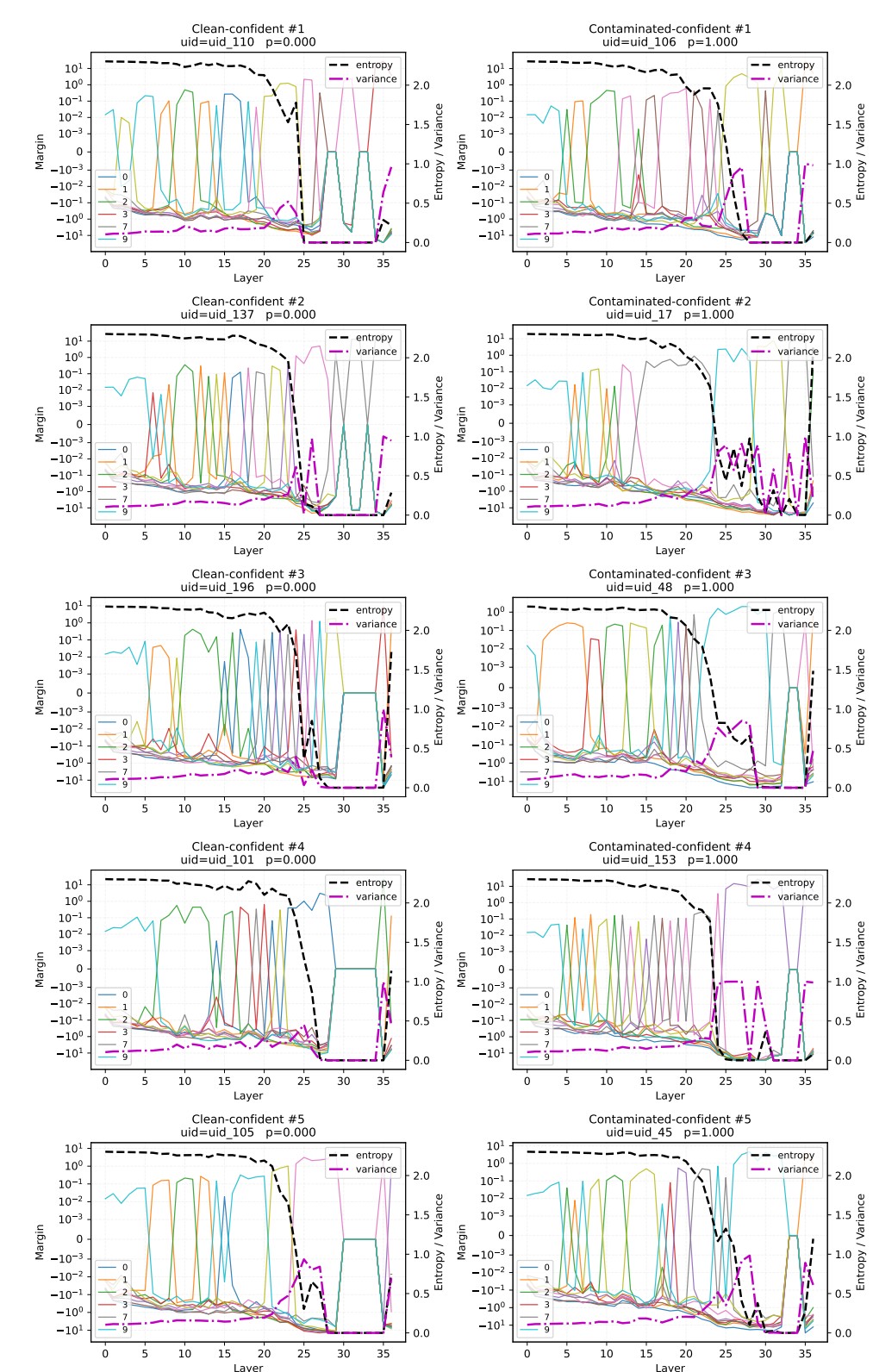

**Figure 11:** Activation trajectories of the 10 most confidently classified samples by the discriminator trained on Qwen3-8B. Clean samples are shown on the left, while contaminated samples are shown on the right.

# E IMPLEMENTATION DETAILS

## E.1 DISCRIMINATOR TRAINING

**Feature construction.** For each problem–answer pair, we tokenize the prompt `''Problem: <q>\nAnswer:''` with the model's native tokenizer and run the model with `output_hidden_states=True`. At every layer $l$, we compute (i) the normalized probability mass over digit tokens $p_l(d)$ for $d \in \{0, \ldots, 9\}$ (token sets collected from the tokenizer vocabulary), (ii) the entropy $H_l$ of $p_l$, and (iii) the maximum probability $\max_d p_l(d)$. We also take first-order differences along depth for each of these streams. Stacking gives a $C \times T$ trajectory with $C = 24$ channels: 10 digit channels $p_l(d)$, plus $H_l$ and $\max p_l$, and their 12 first-order differences. We z-score normalize *per sample, per channel* (NaNs $\to 0$). For sequences with different $T$, we truncate all samples to the minimal $T$ across the batch to align time axes.

**Discriminator architecture.** We use a lightweight 1D CNN (`TinyTSConv`) with three `Conv1d` layers (kernel size 3, channels $C \to 64 \to 128 \to 128$) and ReLU activations, followed by global average pooling and a 2-way linear head. Inputs have shape $[N, C, T]$ with $C = 24$.

**Training protocol.** We perform a stratified split with validation ratio 0.2. Optimization uses AdamW with learning rate $2 \times 10^{-3}$, batch size 32, and cross-entropy loss. To address class imbalance, we apply inverse-frequency class weights $w_k = \dfrac{n_{\text{pos}} + n_{\text{neg}}}{2 \max(n_k, 1)}$. We train for 15 epochs and select the checkpoint that maximizes AUROC on the validation set. All random seeds are fixed to $42$ (Python/NumPy/PyTorch). Training runs on the same device as the backbone model (CPU/GPU), with features stored as `float32`.

## E.2 LORA FINE-TUNING EXPERIMENT

**LoRA fine-tuning.** We fine-tune *Llama-3.1-8B* using LoRA adapters applied to attention and MLP projection layers (`q_proj`, `k_proj`, `v_proj`, `o_proj`, `gate_proj`, `up_proj`, `down_proj`). Training is performed in 4-bit NF4 quantization with double quantization, and all experiments vary the LoRA rank $r$ while setting $\alpha = 2r$ and dropout 0.05. Problems are concatenated and segmented into blocks of 1024 tokens, where the model is trained in the standard causal language modeling objective with cross-entropy loss. We use AdamW with learning rate $1 \times 10^{-4}$, cosine decay, warmup ratio 0.03, batch size 2 per device with gradient accumulation 8, and train for 32 epochs. Gradient checkpointing and TF32 are enabled. All random seeds are fixed to 0 for reproducibility.

