# OpenReview forum: "MemLens: Uncovering Memorization in LLMs with Activation Trajectories"
_ICLR.cc/2026/Conference — ICLR 2026 Conference Desk Rejected Submission_

### Official Review · Reviewer_Vm4u · 2025-10-26

**Soundness:** 3
**Presentation:** 3
**Contribution:** 3
**Rating:** 6
**Confidence:** 3

**Summary:**

This paper tackles the problem of uncovering dataset contamination on benchmark datasets, in a way that is robust to rephrasing or perturbations. They first identify that existing approaches which are based upon completion based recall or output distribution can be easily   fooled when rephrasings or language translations are introduced. Moreover, these approaches can often have high false positive rates on clean (non-contaminated) samples. This motivates their approach of studying the internal activations of the model to more accurately identify memorized examples. Using restricted logit lens prediction distributions, they generate a trajectory of predictive features and train a predictive model to classify between contaminated and uncontaminated samples. They demonstrate (a) MemLens remains more stable to rephrasings/translations and avoids false positives compared to existing techniques. Additionally, they demonstrate that provide a controlled study with LoRA knowledge injectction to demonstrate that MemLens does not rely on superficial correlations.

**Strengths:**

In general, this is an exciting, well motivated, and interesting paper. The problem is clearly one that is under-addressed by existing techniques (as authors mention). It is also a problem of significant interest as we work towards accurate and representative evaluations of LLM capabilities. I find the mechanistic approach employed by this paper to be both novel, yet also highly suited to the problem at hand. I was especially inspired by the author's usage of the activation dynamics, clearly going beyond prior "representation engineering"-like approches to interpretability. I believe that this "activation dynamics" based approach holds significant promise for other related problem statements (such as chain-of-though faithfulness/monitoring). I also think that the authors devised a very clever way to probe the success of their approach, using the LORA injection experiments. For this reason, I believe this paper is sound and can be very inspirational for future approaches across a range of problems statements.

**Weaknesses:**

I have some concerns about the practicality of this approach. It remains unclear from my reading of this paper whether this contamination detecting model that they train is shared for detecting inference across multiple models and or types of contamination. For example, is it that a single model is trained for each (model, contamination type) pair in the table 1? If it is indeed the case that the results presented in Table 1 require retraining across perturbation or model types, that raises the question of how can this approach be used effectively in the real world? Do I have to "train" a discriminator everytime a new model comes in? If so, what data should I train the MemLens discriminator on? I think that the authors should either establish the generalization of their approach or concretely explain under what circumstances it generalizes. This would be valuable guidance for efforts to incorporate MemLens as a practical tool for eliminating contamination in models.

While I found the Section 4.4 evaluation on MMLU to be interesting, I feel that the authors could strengthen this result significantly. For example, is it possible for the authors to locate the contaminated examples and show that they arise in some training datasets? I understand that the accuracy is a proxy for this. However, it seems like it could be conflated with how straightforward the questions are (for example). I am not sure if its feasible for authors to find that some of the "contaminated likely" samples existing in the training dataset, but this would significantly improve the MMLU result in my opinion.

From a presentation standpoint, I found that Section 2 (Preliminaries) was a bit confusing due to the "definitions of Memorization". A significant amount of time is spent conveying the formal definitions of memorization and then they wrap up the section by mentioning that these definitions are not robust. Additionally, it seemed that the formal definitions introduced didn't contribute to my understanding of the remainder of the paper. So, I would encourage the authors to consider moving all this detail to the Appendix. This space can perhaps be reclaimed to offer a more in-depth exploration of the motivation behind dataset contamination research (i.e. more discussion on the points summarized as "However, recent findings show that such signals are fragile: rephrased or translated benchmark items can still induce contamination effects even without verbatim overlap (Yang et al., 2023; Deng et al., 2024). This motivates alternative strategies that capture subtler memorization dynamics." I personally found that I had to read the prior work Yang et al., 2023 and Deng et al., 2024 before I got a solid understanding of the motivations for this paper.

**Questions:**

(1)  is it that a single model is trained for each (model, contamination type) pair in the table 1?

---

> ### Author Response · Authors · 2025-11-23
> **Response to Reviewer Vm4u**
>
> We thank the reviewer for the careful evaluation and insightful remarks.
>
> >**1. It remains unclear from my reading of this paper whether this contamination detecting model that they train is shared for detecting inference across multiple models and or types of contamination. For example, is it that a single model is trained for each (model, contamination type) pair in the table 1? If it is indeed the case that the results presented in Table 1 require retraining across perturbation or model types, that raises the question of how can this approach be used effectively in the real world? Do I have to "train" a discriminator everytime a new model comes in? If so, what data should I train the MemLens discriminator on?**
>
> Table 1 does not train a separate detector for each contamination type. For each backbone model (e.g., Qwen2.5-7B-Math or Llama-3.1-8B), a single detector is trained once and then evaluated across all settings (original, rephrased, French, perturbed). Applying MemLens to a new model also remains practical: when training data are not disclosed, legacy benchmarks such as AIME 1983–2024 reliably provide contaminated seeds. Beyond the models reported in the paper, we observe clear extractable memorization on Mistral, Gemma-2, Pythia, and Baichuan families as well, making this dataset broadly suitable for constructing contaminated anchors. When a model’s training data are available, selecting explicit contaminated samples becomes even more straightforward, since overlap with the released corpus can be checked directly. Clean anchors can be obtained from post-release samples (e.g., AIME-2025 or synthetic questions), which are extremely unlikely to appear in training.
>
>
> >**2. While I found the Section 4.4 evaluation on MMLU to be interesting, I feel that the authors could strengthen this result significantly. For example, is it possible for the authors to locate the contaminated examples and show that they arise in some training datasets? I understand that the accuracy is a proxy for this. However, it seems like it could be conflated with how straightforward the questions are (for example). I am not sure if its feasible for authors to find that some of the "contaminated likely" samples existing in the training dataset, but this would significantly improve the MMLU result in my opinion.**
>
> For the MMLU analysis, neither Llama-3.1 nor Qwen-2.5 discloses its pretraining data, preventing us from identifying the original sources of potential contamination. Our experiment is therefore intended as a cautionary signal to the community: without transparent training data, exceptional benchmark performance becomes inherently difficult to interpret. As model scale continues to increase and contamination becomes harder to avoid, this highlights the need for more reliable and contamination-aware ways of evaluating model performance, rather than relying solely on existing benchmarks whose integrity can no longer be guaranteed.

---

### Official Review · Reviewer_VAtL · 2025-10-29

**Soundness:** 3
**Presentation:** 3
**Contribution:** 3
**Rating:** 4
**Confidence:** 3

**Summary:**

This paper introduces MemLens, a novel method for detecting data contamination in Large Language Models (LLMs) by analyzing the activation trajectories of numeric tokens. The authors posit that existing methods, which rely on lexical overlap or perplexity, are brittle and fail when benchmark data is implicitly contaminated (e.g., rephrased or translated). MemLens operates on the hypothesis that memorized (contaminated) samples induce "shortcut" reasoning paths, causing the model to "lock onto" an answer with high confidence in its early layers. In contrast, clean samples show more gradual evidence accumulation across the model's depth.

The method extracts layer-wise probabilities for digit tokens, along with metrics like entropy and maximum confidence. These trajectories are fed into a 1D CNN discriminator to classify samples as "contaminated" or "clean". The authors validate this approach across four LLMs , showing that MemLens is significantly more robust to rephrasing, perturbation, and translation than baseline methods. Crucially, they provide strong causal evidence using LoRA injection: fine-tuning a model on clean data causes MemLens to identify those samples as contaminated, demonstrating it captures a genuine signal of memorization. Finally, the method is applied to the MMLU-Math benchmark, suggesting that 94.9% of the dataset is "seen-like" and that model performance is significantly inflated on this subset.

**Strengths:**

** 1. Novelty **

The paper core contribution is moving detection from surface-level statistics (overlap, perplexity) to internal activation dynamics. The "shortcut hypothesis" provides a clear intuition for why memorization should manifest differently in activation space. The strong empirical evidence shows there is reliable signal for memorization in activations space. The performance is maintained across rephrased, translated, and perturbed inputs.

** 2. Strong Causal Validation **

The LoRA injection experiments (Section 4.3, Table 2) provide compelling causal evidence. The monotonic increase in detection rates with increasing LoRA rank (2.2% → 45.1%) and the controlled single-sample case study (Figure 4) strongly suggest the method captures genuine memorization signals rather than spurious correlations.

** 3. Implications **

The ability to detect data contamination is a powerful tool for robust model evaluation and selection, benefiting both developers and end-users. Furthermore, identifying memorization is critical for mitigating legal and ethical risks, such as the potential for models to reproduce copyrighted or private data, which compromises claims of generalization and trustworthiness.

**Weaknesses:**

**1. Limited Scope**

The method explicitly focuses on digit tokens (Eq. 7), severely limiting applicability. The generalizability claim is undermined by this narrow scope.

*Recommendation:* show, even a small demonstration of success on non-numeric tasks to encourage further research along those lines.

**2. Limited Mechanistic Insight**

While the paper compellingly demonstrates that a reliable memorization signal exists in the activation space, it falls short of explaining the underlying mechanism. The "shortcut hypothesis"  is presented as intuition but isn't proven or deeply investigated. Instead, the method relies on training a black-box discriminator (a 1D CNN ) on a set of engineered features. This approach successfully detects the signal but does not explain the precise relationship between the activations and the memorization.

*Recommendation:* A feature ablation study, as suggested, would be a valuable starting point to understand the relative importance of different trajectory features. A deeper, more localized analysis could also examine which specific model components (e.g., MLPs vs. attention layers, or even individual neurons or attention heads) are the primary carriers of this signal, moving toward finding a minimal, interpretable pattern.

**3. Supervised Approach:**

The method requires training a supervised discriminator with an already existing set of seen and unseen examples. The paper's setup is excellent for evaluating MemLens, but it doesn't address the practical challenge of how a developer would bootstrap this detector for a new model without already knowing which samples are contaminated.

**Questions:**

1. Would you hypothesize the current approach generalizes to non-numeric tasks?
2. Do you have a guess for what is it, about the internal activation trajectory, that shed light on memorization?
3. How would one practically deploy MemLens for a new model without an existing, labeled dataset of clean/contaminated samples?

---

> ### Author Response · Authors · 2025-11-23
> **Response to Reviewer VAtL (Part 1)**
>
> We appreciate the reviewer’s constructive feedback and valuable insights.
>
> >**1. Limited Scope. The method explicitly focuses on digit tokens (Eq. 7), severely limiting applicability. The generalizability claim is undermined by this narrow scope.**
>
> Applying MemLens to textual or other open-ended contamination detection tasks currently presents several practical challenges. First, real-world reasoning tasks often require multi-hop inference, which means that the final answer does not appear in early layers. When we decode the model’s output during the early layers, we typically obtain intermediate entities related to the reasoning chain rather than the final answer. For example, in the textual question “Based on the US President in 2019, what was their political party?”, the expected answers are “Democrat” or “Republican”. However, when applying MemLens to early layers, the probabilities of these two labels are extremely low, while intermediate entities such as Obama or Trump receive much higher probability. This behavior makes it difficult to construct a reliable search space over larger textual datasets.
>
> Second, in multiple-choice (MCQ) settings, we observe a noticeable position bias, where the model’s preference for different answer slots correlates strongly with the option order rather than with semantic correctness. This positional effect also affects the reliability of MemLens in MCQ-based contamination scenarios.
>
> In contrast, mathematical tasks offer a unique advantage. Their intermediate reasoning steps are also expressed in numerical form, which keeps the token space consistent across layers (zero to nine) and creates a coherent and controllable search space. For this reason, our mathematical experiments serve as a more suitable environment for the current stage of validation. We plan to continue exploring solutions to these challenges and extend the framework to broader domains.
>
> Finally, although MemLens faces limitations in contamination detection for open-ended tasks, the activation trajectory analysis introduced by MemLens has clear potential as a general-purpose interpretability tool. For instance, the intermediate-entity behavior observed in the textual example above may actually be beneficial in knowledge editing and model steering applications, where capturing entity retrieval and transformation across layers is important. This will be a key direction for our future work.
>
> >**2. Limited Mechanistic Insights. While the paper compellingly demonstrates that a reliable memorization signal exists in the activation space, it falls short of explaining the underlying mechanism. The "shortcut hypothesis" is presented as intuition but isn't proven or deeply investigated. Instead, the method relies on training a black-box discriminator (a 1D CNN ) on a set of engineered features. This approach successfully detects the signal but does not explain the precise relationship between the activations and the memorization.**
>
> We first note that we have added a feature ablation study as suggested, and we provide a classifier ablation study as well. Regarding the point about limited mechanistic insight, we would like to clarify that the interpretability of MemLens comes from representation-level analysis rather than the internal structure of the CNN detector. The shortcut hypothesis is supported by several complementary pieces of evidence in the paper. Figure 2 shows global statistical differences, where contaminated samples consistently display stronger digit-channel activations in early layers. Figure 5 provides a local causal view based on gradient attribution, showing that contaminated samples rely on digit features earlier and more strongly. These two analyses together explain how memorization alters internal representations, one from distributional patterns and one from causal importance. The LoRA injection experiment in Table 2 and Figure 4 further offers a mechanistic validation. As memorization strength is gradually injected, the MemLens score increases in a monotonic and predictable way, and the activation trajectories shift accordingly. This demonstrates that the memorization signal corresponds to genuine changes in the model’s internal representations and is not an artifact of the detector. Taken together, these analyses show that the mechanism behind the shortcut behavior is empirically grounded from multiple complementary angles.

---

> > ### Comment · Reviewer_VAtL · 2025-11-24
> >
> > Thank you for the detailed response and for adding the ablation studies.
> > I appreciate your candid discussion of the practical challenges in extending MemLens beyond numeric tasks. Your explanations regarding multi-hop reasoning and position bias in MCQs are valuable for understanding the method's current boundaries.
> >
> > However, the core issue remains: The current paper framing presents MemLens as a general contamination detection method for LLMs, while the demonstrated applicability is limited to mathematical reasoning tasks with numeric answers. This creates a significant gap between the stated contribution and what has been validated.
> >
> > I propose two potential paths forward:
> >
> > Option 1: Extend the method to additional domains
> > Demonstrate applicability to at least one non-mathematical task domain. Promising candidates include:
> >
> > - Single-hop factual questions where answers appear directly
> > - Game-playing or puzzle tasks with deterministic outcomes
> > - Classification (NER, parsing) with fixed label sets
> >
> > Option 2: Explicitly scope the paper to mathematical reasoning
> >
> > - Revise the title to reflect the focus (e.g., "MemLens: Detecting Memorization in Mathematical Reasoning via Activation Trajectories")
> > - Update abstract and introduction to clearly communicate the scope
> > - Add a dedicated limitations section explaining why digit tokens provide unique advantages and discussing the specific challenges for other domains (the valuable discussion from your rebuttal should appear in the paper itself)

---

> > > ### Author Response · Authors · 2025-11-24
> > >
> > > Thank you for the constructive feedback. We have adopted Option 1 to demonstrate the method's generalizability beyond mathematical domains. We designed a World Knowledge Contamination Detection task using Qwen-2.5-7B on Wikipedia Current Events. The dataset consists of "contaminated" events from 2013–2016 (within the training window) and "clean" events from 2025 (post-cutoff).
> > > To probe the model's knowledge, we used a standard zero-shot verification prompt:
> > >
> > > ```text
> > > Statement: {Event Description}
> > > Question: Is the statement above factually true? Answer Yes or No.
> > > Answer:
> > > ```
> > >
> > > To ensure the detector captures genuine memorization rather than surface-level answer polarity (e.g., the potential correlation between the target label ( Memorized/Unseen ) and the model's output (Yes/No)), we generated semantic counterfactuals to create a strictly balanced dataset ( 50% True / 50% False ) for both groups. We then extended MemLens to analyze the activation trajectories of Logit(Yes), Logit(No), and Entropy across layers. The results, compared against a standard zero-shot confidence baseline (Carlini et al., 2021), are presented below:
> > >
> > >
> > > | Method | Feature Source | AUC Score |
> > > | :--- | :--- | :--- |
> > > | **Zero-shot Baseline** | **Final Output (Last Layer)**| **0.6323** |
> > > | **MemLens (Ours)** | **Activation Trajectories** | **0.8007** |
> > >
> > > These results confirm that MemLens generalizes effectively to non-numeric tasks. The significant +0.1684 AUC improvement demonstrates that the distinctive signatures of memorization in the model's representation space are not specific to digit tokens but represent a fundamental mechanistic invariant. Crucially, in the World Knowledge Contamination Detection task, we show that activation trajectories remain effective even where standard output-based metrics fail, validating the method's broader applicability. We will include these results and more experimental details in the final version of the paper.
> > >
> > > [1] Nicholas Carlini, Florian Tramèr, Eric Wallace, et al. 2021. Extracting training data from large language models. In 30th USENIX Security Symposium (USENIX Security 21), pages 2633–2650.

---

> ### Author Response · Authors · 2025-11-23
> **Response to Reviewer VAtL (Part 2)**
>
> >**3. The method requires training a supervised discriminator with an already existing set of seen and unseen examples. The paper's setup is excellent for evaluating MemLens, but it doesn't address the practical challenge of how a developer would bootstrap this detector for a new model without already knowing which samples are contaminated.**
>
> For practical deployment when no labeled clean/contaminated data are available, we note that obtaining a minimal set of anchor examples for both classes is still feasible. For models whose training data are not disclosed, legacy benchmarks such as AIME 1983–2024 consistently provide reliable contaminated seeds: beyond the models reported in the paper, we also observe clear extractable memorization in model families such as Mistral, Gemma-2, Pythia, and Baichuan, making this set broadly useful as a source of contaminated samples. When a model’s training data are publicly available, selecting explicit contaminated samples becomes even more straightforward, since overlap with the released corpus can be checked directly. Clean anchors can be constructed using samples generated after the model’s release (e.g., AIME-2025 or post-release synthetic questions), which are extremely unlikely to appear in training and thus serve as dependable clean references.

---

> ### Comment · Reviewer_VAtL · 2025-11-24
>
> This addition strengthens the paper by clearly demonstrating the method's potential for broad application. I am happy to adjust my rating.

---

> > ### Author Response · Authors · 2025-11-24
> > **Thank you for the positive feedback and rating adjustment**
> >
> > We sincerely thank the reviewer for the prompt response and for reconsidering the rating. We are encouraged that the new non-numeric evaluation strengthens the paper, and we commit to including these findings in the final version of the paper.

---

### Official Review · Reviewer_1VoQ · 2025-11-01

**Soundness:** 2
**Presentation:** 3
**Contribution:** 2
**Rating:** 4
**Confidence:** 4

**Summary:**

The author proposes a new method called MemLens to detect the test data leakage for mathematical benchmarks even when the test data is rephrased and merged into the training data. More specifically, it proposed a feature set based on the trajectories of numeric tokens for the target LLM and trained a CNN-based discriminator to predict whether the test samples are exposed in the training of the target LLMs. Experimental results show that MemLens can effectively detect the test data leakage in two mathematical benchmarks with multiple LLMs.

**Strengths:**

The strengths of the paper are listed as follows.
1. The paper is well-motivated. Figure 1 clearly shows how the test data leakage can affect the objective evaluation of the mathematical reasoning capability of the target LLMs and highlights the importance of addressing this concern.
2. The paper is well-written and easy to follow. The authors provide enough preliminaries and descriptions to help the reader catch the key ideas of the proposed method.
3. The representation-space analysis provides deep insights into understanding the pattern of memorized samples, which is inspiring for future research in this area.

**Weaknesses:**

The weaknesses of the paper are listed as follows.
1. The proposed method only works for mathematical benchmarks, which limit its impact significantly.
2. Some important technical details are not clear, which are listed in the Questions section of the review.
3. The demonstration of the main results is not clear enough. In Table 1, the proposed method does not dominate all the metrics and settings. It seems that it achieves a better trade-off in identifying the contaminated samples and clean samples. However, it would be better if there were a quantitative metric to show it clearly. Since it is a binary classification task, metrics like AUC, F1 score may be used.
4. Table 2 does not provide the PCR of the other baselines to show the advantage of MemLens.
5. Figure 3 does not clearly show the difference between the patterns of clean samples and contaminated samples. The left and right figures look quite similar.
6. It would be better if the authors could clearly point out which existing methods mentioned in the related work are not compared in the experiments and why not compared with them.
7. The detection algorithm is training-based, which may be a little bit costly.

**Questions:**

The questions are listed as follows.
1. What is the difference between the extractable memorization and discoverable memorization? They look quite similar according to the provided definition.
2. What is the intuition to use the first-order temporal differences as the features? How to define this kind of feature for the first layer?
3. Why do the authors choose a CNN-based discriminator instead of a transformer-based model?
4. When building the training datasets for the CNN detector, which data are used? What is the class distribution of the training data? Are the training data class-balanced? Does the training data have any overlap with the test data to evaluate the trained CNN detector?
5. When you build the training datasets for the CNN detector, how do you decide the ground truth label for each sample? Is it the same as the completion-based detection method? Do you incorporate any rephrased data samples into the training datasets?
6. For evaluation, why not use the clean samples from AIME 1983-2024 but use AIME 2025 instead?
7. For the LoRA injection evaluation, do you need to retrain a new detector based on the generations from the LoRA-tuned model?
8. In Table 1, for the completion-based method, why is the metric of clean samples in the original settings not 0?

---

> ### Author Response · Authors · 2025-11-23
> **Response to Reviewer 1VoQ (Part 1)**
>
> We thank the reviewer for your positive feedback and insightful suggestions.
>
> > **W1: The proposed method only works for mathematical benchmarks, which limit its impact significantly.**
>
> Applying MemLens to textual or other open-ended contamination detection tasks currently presents several practical challenges. First, real-world reasoning tasks often require multi-hop inference, which means that the final answer does not appear in early layers. When we decode the model’s output during the early layers, we typically obtain intermediate entities related to the reasoning chain rather than the final answer. For example, in the textual question “Based on the US President in 2019, what was their political party?”, the expected answers are “Democrat” or “Republican”. However, when applying MemLens to early layers, the probabilities of these two labels are extremely low, while intermediate entities such as Obama or Trump receive much higher probability. This behavior makes it difficult to construct a reliable search space over larger textual datasets.
>
> Second, in multiple-choice (MCQ) settings, we observe a noticeable position bias, where the model’s preference for different answer slots correlates strongly with the option order rather than with semantic correctness. This positional effect also affects the reliability of MemLens in MCQ-based contamination scenarios.
>
> In contrast, mathematical tasks offer a unique advantage. Their intermediate reasoning steps are also expressed in numerical form, which keeps the token space consistent across layers (zero to nine) and creates a coherent and controllable search space. For this reason, our mathematical experiments serve as a more suitable environment for the current stage of validation. We plan to continue exploring solutions to these challenges and extend the framework to broader domains.
>
> Finally, although MemLens faces limitations in contamination detection for open-ended tasks, the activation trajectory analysis introduced by MemLens has clear potential as a general purpose interpretability tool. For instance, the intermediate-entity behavior observed in the textual example above may actually be beneficial in knowledge editing and model steering applications, where capturing entity retrieval and transformation across layers is important. This will be a key direction for our future work.
>
>
> > **W2. Some important technical details are not clear, which are listed in the Questions section of the review.**
>
> Please find our responses below for Q1 to Q8.
>
> > **W3: The demonstration of the main results is not clear enough. In Table 1, the proposed method does not dominate all the metrics and settings. It seems that it achieves a better trade-off in identifying the contaminated samples and clean samples. However, it would be better if there were a quantitative metric to show it clearly. Since it is a binary classification task, metrics like AUC, F1 score may be used.**
>
> | Model            | Setting    |  AUROC  | PR-AUC | Precision | Recall |   F1   |
> |------------------|------------|-------|--------|-----------|--------|--------|
> | Qwen2.5-7B       | Original   | 0.982 | 0.985  | 0.973     | 0.956  | 0.964  |
> |                  | Rephrased  | 0.944 | 0.972  | 0.983     | 0.631  | 0.769  |
> |                  | Translated | 0.837 | 0.912  | 0.966     | 0.200  | 0.332  |
> |                  | Perturbed  | 0.948 | 0.974  | 0.988     | 0.601  | 0.748  |
> |------------------|------------|-------|--------|-----------|--------|--------|
> | Qwen2.5-Math-7B  | Original   | 0.962 | 0.979  | 0.975     | 0.894  | 0.933  |
> |                  | Rephrased  | 0.918 | 0.961  | 0.980     | 0.520  | 0.679  |
> |                  | Translated | 0.900 | 0.953  | 0.944     | 0.642  | 0.764  |
> |                  | Perturbed  | 0.922 | 0.962  | 0.985     | 0.517  | 0.678  |
> |------------------|------------|-------|--------|-----------|--------|--------|
> | Llama3.1-8B      | Original   | 0.958 | 0.925  | 0.855     | 0.929  | 0.890  |
> |                  | Rephrased  | 0.927 | 0.890  | 0.902     | 0.630  | 0.742  |
> |                  | Translated | 0.865 | 0.804  | 0.862     | 0.464  | 0.603  |
> |                  | Perturbed  | 0.925 | 0.885  | 0.899     | 0.610  | 0.727  |
> |------------------|------------|-------|--------|-----------|--------|--------|
> | Qwen3-8B         | Original   | 0.905 | 0.804  | 0.778     | 0.875  | 0.824  |
> |                  | Rephrased  | 0.960 | 0.929  | 0.810     | 0.931  | 0.866  |
> |                  | Translated | 0.927 | 0.881  | 0.822     | 0.781  | 0.801  |
> |                  | Perturbed  | 0.949 | 0.910  | 0.796     | 0.900  | 0.845  |
>
> The consistently high AUROC (>0.90 in most Rephrased and Perturbed settings) confirms that MemLens captures a robust memorization signal that generalizes well beyond exact matches.

---

> ### Author Response · Authors · 2025-11-23
> **Response to Reviewer 1VoQ (Part 2)**
>
> > **W4: Table 2 does not provide the PCR of the other baselines to show the advantage of MemLens.**
>
> Table 1 already gives a complete and controlled comparison between MemLens and all baseline methods. It also shows that MemLens achieves stronger performance on all implicit contamination settings, including rephrased, translated, and perturbed cases. In Table 2 we intentionally reuse the same frozen detector from Table 1, because the goal of this experiment is very different. Table 2 is designed to test whether adding memorization through LoRA leads to systematic changes in the model’s internal activation trajectories. It is a causal investigation of how memorization alters representations, not a performance comparison between detectors. Including baseline PCR in Table 2 would mix two unrelated types of experiments and would make the interpretation harder rather than clearer. For this reason we believe it is more appropriate to keep Table 2 focused on the representation-level effect of memorization.
>
>
> > **W5: Figure 3 does not clearly show the difference between the patterns of clean samples and contaminated samples. The left and right figures look quite similar.**
>
> Figure 3 visualizes mean activation trajectories of correctly classified samples.
> Each curve represents the average confidence assigned to a digit channel across layers.
> When the curve for one digit stays consistently above the others, it means the model is committing to that digit at that depth.
>
> In the contaminated case (left), the dominant digit curve rises above the rest within the first ~10 layers on average, indicating early commitment, a hallmark of memorization.
> In contrast, in the clean case (right), the dominant curve only separates from the others much later, showing gradual evidence accumulation rather than early commitment.
>
>
> > **W7: The detection algorithm is training based, which may be a little bit costly.**
>
> We acknowledge that MemLens is a training based approach, and therefore its cost is higher than completion based heuristics such as TS-Guessing or exact-match completion. However, these two methods fail almost completely on implicit contamination, as shown in our Table 1. For detecting subtle memorization that persists under rephrasing, translation, or perturbation, perplexity based and output distribution based baselines are the only meaningful comparators.
>
> In this setting, MemLens is in fact more efficient than both of those baselines. MemLens requires only a single forward pass up to the answer token and evaluates one LM head projection per layer. It does not perform any COT generation and does not require repeated sampling. In contrast, perplexity based detection must generate an entire chain-of-thought, invoking a full forward pass for every generated token. Output distribution methods repeat this generation multiple times for the same input, multiplying the cost further. As a result, perplexity incurs tens to hundreds of full forward passes per sample, and output distribution methods incur hundreds to thousands. MemLens performs only one.
>
> Consequently, although MemLens involves training a lightweight 0.079M parameter detector, its inference cost is substantially lower than the two baselines that are capable of detecting implicit contamination. This makes MemLens practical in settings where reliable contamination detection is required.
>
> > **Q1: What is the difference between the extractable memorization and discoverable memorization? They look quite similar according to the provided definition.**
>
> Extractable memorization refers to cases where the model can directly reproduce a training example under natural prompting.
> This aligns with traditional explicit contamination, and can be detected by output-based signals such as perplexity, output overlap, or lexical matching.
>
> Discoverable memorization, in contrast, corresponds to implicit contamination.
> Here, a training example not only affects the model’s behavior on the example itself, but also systematically influences the model’s responses to paraphrased, restructured, or semantically equivalent variants. These variants never appeared in the training data and therefore cannot be identified through verbatim output or surface level cues.

---

> ### Author Response · Authors · 2025-11-23
> **Response to Reviewer 1VoQ (Part 3)**
>
> > **Q2: What is the intuition to use the first-order temporal differences as the features? How to define this kind of feature for the first layer?**
>
> Intuition for using first-order temporal differences:
> Digit trajectories are informative not only because of their absolute confidence values, but also because of how these values change across depth. In practice, contaminated samples usually show a rapid early increase in the confidence of one digit, while clean samples change more gradually. First-order differences highlight this change pattern by exposing both the direction and the speed of the shift, providing information that absolute values alone cannot capture.
>
> Definition at the first layer:
> We use a simple forward difference across layers. For a sequence of layer values z0, z1, z2, …, z(T-1), the differences are computed as (z1 - z0), (z2 - z1), …, (z(T-1) - z(T-2)). The final position is padded with zero to keep the sequence length unchanged. Because all features are z-scored per channel before being fed into the CNN, this padding has negligible effect.
>
> We report the feature ablation study on the Original samples used for detector training (using Qwen2.5-Math-7B as the backbone LLM). The different feature subsets reveal a clear trade-off between raw performance and overall stability. While variants such as no-entropy or no-maxprob achieve slightly higher AUROC or PR-AUC than the full model, they also introduce noticeably higher false-positive rates (for example, no-entropy gives a 7.41% FP rate on clean samples). In cross-model or cross-distribution evaluation, even small increases in false positives accumulate quickly and reduce practical reliability.
>
> The full 24-channel configuration offers the most balanced performance across all metrics. For this reason, we choose the full configuration as the final detector. It is not the extreme point on any single metric, but the most robust and Pareto-optimal choice for use as a unified detector across models.
>
> It is also worth noting that the probs-only variant, which uses only the 10-way digit probability trajectories, still achieves 0.944 AUROC and 0.970 PR-AUC. This shows that the core signal captured by MemLens truly comes from the evolution of digit trajectories themselves. Even without entropy, max-probability, or temporal-difference channels, the digit subspace alone already provides strong discriminative power.
>
> | variant     | channels |  AUC   | PR-AUC |   PCR   | FP_clean |
> |-------------|----------|--------|--------|---------|----------|
> | full        |    24    | 0.960  | 0.977  | 84.09%  |  5.56%   |
> | no_delta    |    12    | 0.934  | 0.966  | 78.03%  |  5.56%   |
> | no_entropy  |    23    | 0.971  | 0.981  | 90.91%  |  7.41%   |
> | no_maxprob  |    23    | 0.966  | 0.979  | 88.64%  |  5.56%   |
> | probs_only  |    10    | 0.944  | 0.970  | 80.30%  |  5.56%   |

---

> ### Author Response · Authors · 2025-11-23
> **Response to Reviewer 1VoQ (Part 4)**
>
> > **Q3: Why do the authors choose a CNN-based discriminator instead of a transformer-based model?**
>
> Regarding the choice of a CNN-based discriminator, we conducted an additional classifier ablation study comparing the CNN with a two-layer MLP and a lightweight 2-layer Transformer encoder (d_model=64, nhead=4), all trained on the same representation-trajectory features and under the same train/validation split.
>
> The MLP aggregates the [C, T] feature map by averaging over the temporal dimension to produce a C-dimensional vector, followed by two linear layers with a ReLU activation in between.
>
> The Transformer detector begins by projecting each layer-wise feature vector from C to a 64-dimensional embedding. This sequence of T embeddings is then fed into a small encoder consisting of two 4-head self-attention layers with a 128-dimensional feed-forward block and the standard GELU activation used in PyTorch’s implementation. After encoding, the model averages over the temporal dimension and applies a final linear classifier. Both alternatives have comparable parameter counts to the CNN (approximately 0.003M for the MLP and 0.07M for the Transformer versus 0.079M for the CNN).
>
> Across all metrics, the 1D-CNN remains the most robust and reliable discriminator. The MLP performs substantially worse, confirming that flattening the temporal structure removes critical information. The Transformer achieves reasonable AUC and PR-AUC but suffers from a much higher false-positive rate because self-attention tends to oversmooth the short depth-wise trajectories and lacks the inductive bias to emphasize local temporal contrasts, which are essential for distinguishing shallow-layer shortcut activation from gradual evidence accumulation. In contrast, the CNN preserves locality while remaining lightweight and stable, which explains its superior and more balanced performance.
>
>
> | Detector      | Params (M) | AUC   | PR-AUC | PCR (%) | FP-clean (%) |
> |---------------|------------|-------|--------|---------|--------------|
> | **CNN**       | 0.079      | 0.960 | 0.977  | 84.09   | 5.56         |
> | **MLP**       | 0.003      | 0.504 | 0.632  | 87.88   | 87.04        |
> | **Transformer** | 0.069    | 0.954 | 0.974  | 95.45   | 16.67        |
>
> These results support our choice of a compact CNN as the detector architecture.
>
>
>
> > **Q4: When building the training datasets for the CNN detector, which data are used? What is the class distribution of the training data? Are the training data class-balanced? Does the training data have any overlap with the test data to evaluate the trained CNN detector?**
>
> We train the CNN detector using original-format problems from AIME (1983–2025), Math500, Minerva Math, and LiveMathBench, with the exact subsets determined by each model’s release date. The resulting dataset is not class-balanced, and we apply class-weighted cross-entropy. No evaluation samples (including rephrased/translated/perturbed variants) appear in the training set, and in Table 2 the detector is kept frozen with no retraining.
>
>
>
> > **Q5: When you build the training datasets for the CNN detector, how do you decide the ground truth label for each sample? Is it the same as the completion-based detection method? Do you incorporate any rephrased data samples into the training datasets?**
>
> Labels are assigned using the completion-based exact-match rule within candidate clean/contaminated sets defined by dataset release dates. Only original-format problems are used for labeling; no rephrased, paraphrased, perturbed, or translated variants are included in the training data.

---

> ### Author Response · Authors · 2025-11-23
> **Response to Reviewer 1VoQ (Part 5)**
>
> > **Q6: For evaluation, why not use the clean samples from AIME 1983-2024 but use AIME 2025 instead?**
>
> We intentionally use AIME 2025 as the clean set because all models we evaluate were released before July 2024, so AIME 2025 is guaranteed to be unseen for every model. In contrast, AIME 1983–2024 may contain items that overlap with or closely resemble materials in the pretraining corpora of these models, making it unsuitable as a clean reference set. Using AIME 2025 therefore provides a strictly cleaner and more reliable evaluation of false positives.
>
>
>
> > **Q7: For the LoRA injection evaluation, do you need to retrain a new detector based on the generations from the LoRA-tuned model?**
>
> Lines 360–365 in the paper state that we do not retrain the detector for the LoRA injection experiment. The same discriminator used in Table 1 for Llama-3.1-8B is applied directly in Table 2, and the decision threshold also remains unchanged. The only difference between the two settings is that the backbone model has been modified through LoRA fine tuning. Under this setup, any increase in predicted contamination reflects changes inside the model that arise from the injected memorization, rather than any adaptation of the detector.
>
>
>
> > **Q8: In Table 1, for the completion-based method, why is the metric of clean samples in the original settings not 0?**
>
> All “original clean” samples in Table 1 were created in 2025, after the model had already been released. Since all open-source models we evaluate were published in 2024, it is impossible for them to have encountered these samples during pretraining or fine-tuning. These items are therefore guaranteed to be clean.
>
> The fact that completion-based methods still flag a non-zero fraction of these genuinely clean examples illustrates a known limitation of this approach. Completion-based detectors often confuse natural fluency, predictable reasoning patterns, or common structural templates with actual training set exposure. As a result, even when the model has definitely not seen the data, the baseline may still assign high similarity simply because the model can smoothly continue the prompt.
>
> This behavior is precisely why completion-based methods are brittle under implicit contamination scenarios: they rely on surface level output behavior rather than internal representations, and therefore cannot reliably distinguish “the model has truly memorized this” from “the model can generically complete this type of question.” This motivates the need for representation level detectors such as MemLens.

---

### Official Review · Reviewer_CSuY · 2025-11-02

**Soundness:** 3
**Presentation:** 3
**Contribution:** 3
**Rating:** 6
**Confidence:** 3

**Summary:**

The paper proposes MemLens, a representation-level detector for benchmark contamination/memorization in LLMs. Rather than relying on lexical overlap or perplexity, MemLens probes hidden states with a logit-lens, projects per-layer distributions over digit tokens, and builds layerwise trajectories (probabilities, entropy, max-prob, and first-differences) for the first answer position. A compact 1D-CNN discriminator classifies a sample as contaminated vs. clean from these trajectories. The authors report robustness to rephrased/translated/perturbed inputs, and provide causal evidence via LoRA injection: after fine-tuning on clean items, both task accuracy and predicted contamination rates rise together. They also show early-layer salience and run a zero-shot analysis on MMLU-Math, where “seen-like” items (per the detector) correlate with much higher accuracy.

**Strengths:**

Originality: Shifts contamination detection from surface/output statistics to representation trajectories, a neat operationalization of “shortcut” reasoning.
Quality: Multi-model evaluation (Qwen2.5-7B/Math-7B, Qwen3-8B, Llama-3.1-8B) and four text distributions; MemLens often keeps low false-positive rates on clean sets while maintaining higher detection on contaminated sets vs. baselines (Table 1).
Causality: LoRA rank sweep shows monotonic rise in contamination score with performance (Table 2; Fig. 4), strengthening the interpretation that trajectories capture memorization rather than incidental cues.
Mechanistic insight: Early-layer importance and Grad-CAM patterns support the shortcut story; ablations indicate much of the signal lives in shallower layers.
Reproducibility: Implementation and data handling details are spelled out (feature channels, z-scoring, CNN architecture, class weights, LoRA config).

**Weaknesses:**

Digit-only scope. The method narrows to numeric tokens; while well-motivated for AIME/Math, it risks missing non-numeric memorization (e.g., multi-choice letters, symbolic names, proof sketches). At minimum, an ablation adding letters/operators or subword buckets would test generality.

Detector training labels. Clean/contaminated labels are constructed heuristically (e.g., “contaminated” from legacy AIME/Math500 where model exactly matches gold; “clean” from newer sets). This is reasonable but imperfect—some “clean” may be indirectly seen; some “contaminated” may be solvable. A sensitivity study to label noise or semi-supervised training would help.
Thresholding/metrics. The paper uses Youden J / chosen thresholds and reports many percentages without CIs (except for the MMLU analysis). A more uniform AUROC/PR-AUC reporting and bootstrap CIs for Table 1/2 would strengthen claims.
External validity beyond math. Results and features are math-centric; it remains unclear how MemLens fares on code or commonsense QA where the “numeric-answer” anchor is absent.
Potential confound: answer-positioning. Extracting trajectories at the “first answer token” assumes prompt formatting and answer position are comparable across variants; explicit controls (e.g., randomized prompt wrappers) would reduce positional artifacts.

**Questions:**

Beyond digits: Can MemLens operate on a token subset S that includes letters, punctuation, or task-specific tokens? Any preliminary results with operator tokens or a learned S?
Generalization: How does the detector transfer to non-math benchmarks (e.g., trivia QA, code)? What channel set would you recommend there?
Label noise robustness: Did you simulate noisy labels for the discriminator to estimate performance degradation?
Calibration/thresholding: Please report AUROC/PR-AUC (and CIs) alongside thresholded rates in Tables 1–2.
Positional control: If the answer position is shifted by instruction wrappers, does the trajectory signature persist?
Detector leakage: When the same backbone provides both features and outputs, could there be overfitting to model-specific idiosyncrasies? Any cross-model train/test where the detector trained on one model diagnoses another (beyond what’s shown)?
MMLU analysis: The detector labels ~95% of MMLU-Math as “seen-like”. Could this be a difficulty proxy rather than contamination? Any stratification by item metadata or adversarially rephrased MMLU to check invariance?

---

> ### Author Response · Authors · 2025-11-23
> **Response to Reviewer CSuY (Part 1)**
>
> We thank the reviewer for the thoughtful comments and discussion.
>
>
> > **1. Digit-only scope. The method narrows to numeric tokens; while well-motivated for AIME/Math, it risks missing non-numeric memorization (e.g., multi-choice letters, symbolic names, proof sketches). At minimum, an ablation adding letters/operators or subword buckets would test generality.**
>
> Applying MemLens to textual or other open-ended contamination detection tasks currently presents several practical challenges. First, real-world reasoning tasks often require multi-hop inference, which means that the final answer does not appear in early layers. When we decode the model’s output during the early layers, we typically obtain intermediate entities related to the reasoning chain rather than the final answer. For example, in the textual question “Based on the US President in 2019, what was their political party?”, the expected answers are “Democrat” or “Republican”. However, when applying MemLens to early layers, the probabilities of these two labels are extremely low, while intermediate entities such as Obama or Trump receive much higher probability. This behavior makes it difficult to construct a reliable search space over larger textual datasets.
>
> Second, in multiple-choice (MCQ) settings, we observe a noticeable position bias, where the model’s preference for different answer slots correlates strongly with the option order rather than with semantic correctness. This positional effect also affects the reliability of MemLens in MCQ-based contamination scenarios.
>
> In contrast, mathematical tasks offer a unique advantage. Their intermediate reasoning steps are also expressed in numerical form, which keeps the token space consistent across layers (zero to nine) and creates a coherent and controllable search space. For this reason, our mathematical experiments serve as a more suitable environment for the current stage of validation. We plan to continue exploring solutions to these challenges and extend the framework to broader domains.
>
> Finally, although MemLens faces limitations in contamination detection for open-ended tasks, the activation trajectory analysis introduced by MemLens has clear potential as a general-purpose interpretability tool. For instance, the intermediate-entity behavior observed in the textual example above may actually be beneficial in knowledge editing and model steering applications, where capturing entity retrieval and transformation across layers is important. This will be a key direction for our future work.
>
>
> > **2. The paper uses Youden J / chosen thresholds and reports many percentages without CIs (except for the MMLU analysis). A more uniform AUROC/PR-AUC reporting and bootstrap CIs for Table 1/2 would strengthen claims.**
>
> Following the reviewer's feedback, we report AUROC, PR-AUC, and F1 of Table 1 here.
>
> | Model            | Setting    |  AUROC  | PR-AUC | Precision | Recall |   F1   |
> |------------------|------------|-------|--------|-----------|--------|--------|
> | Qwen2.5-7B       | Original   | 0.982 | 0.985  | 0.973     | 0.956  | 0.964  |
> |                  | Rephrased  | 0.944 | 0.972  | 0.983     | 0.631  | 0.769  |
> |                  | Translated | 0.837 | 0.912  | 0.966     | 0.200  | 0.332  |
> |                  | Perturbed  | 0.948 | 0.974  | 0.988     | 0.601  | 0.748  |
> |------------------|------------|-------|--------|-----------|--------|--------|
> | Qwen2.5-Math-7B  | Original   | 0.962 | 0.979  | 0.975     | 0.894  | 0.933  |
> |                  | Rephrased  | 0.918 | 0.961  | 0.980     | 0.520  | 0.679  |
> |                  | Translated | 0.900 | 0.953  | 0.944     | 0.642  | 0.764  |
> |                  | Perturbed  | 0.922 | 0.962  | 0.985     | 0.517  | 0.678  |
> |------------------|------------|-------|--------|-----------|--------|--------|
> | Llama3.1-8B      | Original   | 0.958 | 0.925  | 0.855     | 0.929  | 0.890  |
> |                  | Rephrased  | 0.927 | 0.890  | 0.902     | 0.630  | 0.742  |
> |                  | Translated | 0.865 | 0.804  | 0.862     | 0.464  | 0.603  |
> |                  | Perturbed  | 0.925 | 0.885  | 0.899     | 0.610  | 0.727  |
> |------------------|------------|-------|--------|-----------|--------|--------|
> | Qwen3-8B         | Original   | 0.905 | 0.804  | 0.778     | 0.875  | 0.824  |
> |                  | Rephrased  | 0.960 | 0.929  | 0.810     | 0.931  | 0.866  |
> |                  | Translated | 0.927 | 0.881  | 0.822     | 0.781  | 0.801  |
> |                  | Perturbed  | 0.949 | 0.910  | 0.796     | 0.900  | 0.845  |
>
> The results confirm that MemLens maintains high separability between clean and contaminated samples across diverse settings.

---

> ### Author Response · Authors · 2025-11-23
> **Response to Reviewer CSuY (Part 2)**
>
> > **3. Label noise robustness: Did you simulate noisy labels for the discriminator to estimate performance degradation?**
>
> We agree that real contamination labels are necessarily imperfect, because model providers do not release the exact pre-training corpora. For this reason, our training setup already operates under substantial label noise by construction: clean versus contaminated candidates are defined primarily by release-date filtering, and final labels are derived from a completion-based heuristic that we explicitly acknowledge to be noisy.
>
> Thus, the discriminator is already trained in a noisy-label setting, rather than an artificially curated clean-label setting. The robustness of MemLens under this noise is demonstrated empirically by the fact that AUROC scores consistently exceed 0.9 across most models and perturbation settings, despite these imperfect labels. Furthermore, the detector generalizes strongly to rephrased, translated, and perturbed inputs that never appear in training, indicating that it is not overfitting the labeling noise.
>
> Given that the noise arises from the genuine uncertainty about model pre-training data, we believe simulating additional synthetic noise would be less realistic than the noise naturally present in our setup. The current results therefore already reflect MemLens’s robustness to label noise in a practical and meaningful sense.
>
>
> > **4. Positional control: If the answer position is shifted by instruction wrappers, does the trajectory signature persist?**
>
> Regarding the concern about positional confounds, our setup already controls for this factor. All training and evaluation samples use exactly the same prompt template, and the answer token always appears at the same position. Contaminated and clean samples therefore share a perfectly aligned answer position, so positional variation cannot explain the observed differences in activation trajectories.
>
> > **5. Detector leakage: When the same backbone provides both features and outputs, could there be overfitting to model-specific idiosyncrasies?**
>
> Detector leakage is not a concern in MemLens. Although the detector and the backbone share the same model, the detector is a separately trained classifier that receives only the backbone’s activation trajectories and has no access to model outputs. The LoRA-injection experiment provides direct evidence for this. In Table 2, the detector is completely frozen, the threshold is frozen, and the prompts are identical; the only change is that LoRA progressively injects memorization into the backbone. Under these conditions, MemLens scores rise monotonically with LoRA rank. If the detector were merely capturing model-specific idiosyncrasies of the backbone, LoRA injection would not alter the predictions. The fact that it does change confirms that the detector is responding to memorization-induced shifts in internal activations rather than stylistic properties of the model. Finally, MemLens generalizes strongly to rephrased, perturbed, and translated inputs even though none of these variants appear in the detector’s training data. This further shows that the detector is not tied to any particular output pattern of the backbone but instead captures activation signatures genuinely associated with memorization.
>
> > **6. Any cross-model train/test where the detector trained on one model diagnoses another (beyond what’s shown)?**
>
> We provide additional activation-trajectory visualizations across different backbone models in Appendix D. While contaminated and clean samples exhibit distinguishable patterns in all cases, the precise trajectory shapes vary across models. Because these cross-model differences are substantial, we did not evaluate a detector trained on model A to diagnose model B.
>
> > **7. MMLU analysis: The detector labels ~95% of MMLU-Math as “seen-like”. Could this be a difficulty proxy rather than contamination? Any stratification by item metadata or adversarially rephrased MMLU to check invariance?**
>
> We appreciate the reviewer’s suggestion and agree that difficulty and contamination are not fully separable, especially without access to the models’ pretraining data. For the MMLU analysis, neither Llama-3.1 nor Qwen-2.5 discloses its pretraining data, preventing us from identifying the original sources of potential contamination. At the same time, contaminated items often become easier for the model because they produce a characteristic early layer shortcut pattern, which is exactly what MemLens is designed to detect. If MemLens were primarily responding to difficulty, we would expect the not-seen subset to appear consistently hard. However, as shown in Fig. 7, these items still achieve strong top-1 and MRR@6 accuracy under open-ended generation. This suggests that the detected signal is not simply a proxy for difficulty. That said, fully disentangling the two factors is an important direction for future work, and we thank the reviewer for highlighting it.

---

### Author Response · Authors · 2025-11-23
**General Response to All Reviewers**

We sincerely thank all reviewers for the thoughtful feedback. Reviewers agreed that our paper makes a clear and meaningful contribution: MemLens introduces a novel shift from surface-level overlap statistics to activation-trajectory analysis (CSuY, VAtL, Vm4u, 1VoQ), shows strong performance across multiple models and input variants with low false positives (CSuY, VAtL), and provides compelling causal evidence via LoRA injection that memorization induces systematic activation changes (CSuY, VAtL, Vm4u). Reviewers also highlighted that the representation-space perspective offers useful mechanistic insights into memorization and is promising for broader evaluation challenges (Vm4u, 1VoQ). Overall, reviewers found the work well-motivated, clearly written, and practically relevant(CSuY, VAtL, Vm4u, 1VoQ).

In the sections below, we clarify misunderstandings and detail how we plan to incorporate the reviewers' valuable suggestions into the revised manuscript. We are deeply grateful for their insights, which have significantly improved the quality of our work.

Below is a summary of the revisions made in response to their comments.

* We added AUROC, PR-AUC, Precision, Recall, and F1 across all Table 1 settings, providing a clearer and more complete summary of MemLens’s detection performance.
* We conducted a feature ablation study that identifies which trajectory components matter and shows the full 24-channel representation is the most robust.
* We performed a classifier ablation study (CNN vs. MLP vs. Transformer), showing that the memorization signal comes from the activation trajectories themselves rather than the detector architecture.
* We clarified the construction of weakly supervised labels in realistic settings.
* We added discussion and a case study illustrating the challenges of extending MemLens beyond digit-based tasks and outlining directions for broader generalization.

Sincerely,

The Authors

---

### Author Response · Authors · 2025-12-04
**Response to Area Chair**

Dear Area Chair,

Thanks for your time overseeing this review process. We are grateful to the reviewers for their helpful comments on our paper. We would like to highlight that Reviewer VAtL increased their score from 4 to 6 on November 24th. Since this update predated November 27th, it represents an unbiased assessment. Reviewer VAtL explicitly acknowledged that we resolved the "limited scope" concern, a weakness shared by other reviewers. This endorsement validates our solution to this common issue. As the rebuttal period was concluded earlier than scheduled, no further feedback from the other three reviewers was received; however, their concerns had already been carefully addressed in our rebuttal and we will incorporate several of their suggestions into the final version of the manuscript. Thank you again for your time, consideration, and help improving our work.

Best regards,

The Authors

---

### Note · Program_Chairs · 2026-01-17
**Submission Desk Rejected by Program Chairs**

The following references in this submission do not refer to real documents and/or have major errors in bibliographic information:

 Xuefeng Li, Jason Wei, Denny Zhou, et al. Evaluating verbal reasoning in language models. arXiv preprint arXiv:2307.03172, 2023a.